# Evolution of atmospheric age of particles and its implications for the formation of a severe haze event in eastern China

Xiaodong Xie[1], Jianlin Hu[1*], Momei Qin[1], Song Guo[2], Min Hu[2], Dongsheng Ji[3], Hongli Wang[4], Shengrong Lou[4], Cheng Huang[4], Chong Liu[5], Hongliang Zhang[6], Qi Ying[7], Hong Liao[1], Yuanhang Zhang[2]

[1] Jiangsu Key Laboratory of Atmospheric Environment Monitoring and Pollution Control, Collaborative Innovation Center of Atmospheric Environment and Equipment Technology, Nanjing University of Information Science & Technology, Nanjing 210044, China
[2] State Key Joint Laboratory of Environmental Simulation and Pollution Control, College of Environmental Sciences and Engineering, Peking University, Beijing, 100871, China
[3] State Key Laboratory of Atmospheric Boundary Layer Physics and Atmospheric Chemistry, Institute of Atmospheric Physics, Chinese Academy of Sciences, Beijing, 100191, China
[4] State Environmental Protection Key Laboratory of Formation and Prevention of Urban Air Pollution Complex, Shanghai Academy of Environmental Sciences, Shanghai, 200233, China
[5] CMA Earth System Modeling and Prediction Centre, State Key Laboratory of Severe Weather, China Meteorological Administration (CMA), Beijing 100081, China
[6] Department of Environmental Science and Engineering, Fudan University, Shanghai 200438, China
[7] Zachry Department of Civil Engineering, Texas A&M University, College Station, Texas 77843, USA

**Corresponding author*: Jianlin Hu (jianlinhu@nuist.edu.cn)

**Abstract**

Atmospheric age reflects how long particles have been suspended in the atmosphere, which is closely associated with the evolution of air pollutants. Severe regional haze events occur frequently in China, influencing air quality, human health, and regional climate. Previous studies have explored the characteristics of mass concentrations and compositions of fine particulate matter ($PM_{2.5}$) during haze events, but the evolution of atmospheric age remains unclear. In this study, the age-resolved UCD/CIT model was developed and applied to simulate the concentration and age distribution of $PM_{2.5}$ during a severe regional haze episode in eastern China. The results indicated that $PM_{2.5}$ concentrations in the North China Plain (NCP) gradually accumulated due to stagnant weather conditions at the beginning stage of the haze event. Accordingly, the atmospheric age of elemental carbon (EC), primary organic aerosol (POA), sulfate ($SO_4^{2-}$), and secondary organic aerosol (SOA) gradually increased. The subsequent $PM_{2.5}$ concentration growth was driven by the local chemical formation of nitrate ($NO_3^-$) under high relative humidity. The newly formed $NO_3^-$ particles led to a decrease in the mean atmospheric age of the $NO_3^-$ particles. During the regional transport stage, aged particles from the NCP moved to the downwind Yangtze River Delta (YRD) region, leading to a sharp increase in $PM_{2.5}$ concentrations and the average age of EC, POA, $SO_4^{2-}$, and SOA. In contrast, the average age of $NO_3^-$ and ammonium remained unchanged or even slightly decreased due to continuous local formation in the YRD region. Different evolution of the atmospheric age among these components provides a unique perspective on the formation of $PM_{2.5}$ components during the regional haze event. The information can also be used for designing effective control strategies for different components of $PM_{2.5}$.

**Keywords:** Atmospheric age; Haze event; Regional transport; Chemical transport model; Eastern China

## 1. Introduction

Haze pollution is a chronic environmental issue in China, influencing atmospheric visibility (Li et al., 2019; Pui et al., 2014), human health (Wang et al., 2021a; Lelieveld et al., 2015; Cohen et al., 2017), ecosystem (Xie et al., 2020; Gu et al., 2002; Cirino et al., 2014), and climate (Ramanathan et al., 2001; Seinfeld et al., 2016; IPCC, 2021). Fine particulate matter, also known as $PM_{2.5}$, the major pollutant during haze days, is either directly emitted into the atmosphere or formed from precursor gases through chemical processes. Although the annual mean $PM_{2.5}$ concentrations in Chinese megacities have been substantially reduced in recent years because of the strict emission control measures (Wang et al., 2019; Zhang et al., 2019b), severe regional haze pollution ($PM_{2.5}$ > 150 μg $m^{-3}$) still frequently occurs in densely populated regions, such as the North China Plain (NCP) and the Yangtze River delta (YRD) (An et al., 2019).

Intensive pollutant emissions and unfavorable meteorology are two key factors controlling haze formation. NCP and YRD are two major city clusters in eastern China with intensive anthropogenic emissions. Previous studies have revealed that severe winter haze events in the NCP were initialized by the accumulation of local emissions under stable weather conditions and further deteriorated by rapid secondary formation (An et al., 2019; Zheng et al., 2015b). Polluted air masses in the NCP are rapidly eliminated by the strong prevailing northwesterly wind and moved to downwind YRD regions (Wang et al., 2021c). During the long-range transport, freshly emitted particles gradually age and mix with secondary inorganic and organic species, further influencing regional climate and air quality through aerosol-planetary boundary layer (PBL) interaction (Huang et al., 2020; Zhang et al., 2021).

The atmospheric age of an air pollutant, defined as the time since it is emitted or formed, provides a unique perspective on the evolution of pollutants in the atmosphere (Wagstrom and Pandis, 2009; Ying et al., 2021; Zhang et al., 2019a). Unlike the lifetime or residence time of pollutants, atmospheric age refers to the time that a single particle remains in the atmosphere at a given location and time, which can better reflect its instantaneous physical and chemical

properties (Chen et al., 2017c). However, measuring and calculating the atmospheric age of air pollutants is difficult because of their chemical nonlinearity and process complexity. Previous studies have attempted to track particle age distributions by adding tracers in Lagrangian trajectory models such as FLEXPART (Stohl et al., 2003). However, due to simplified chemistry, this method cannot accurately determine the age distributions of secondary species. Some other studies estimated the photochemical age of an air mass using the ratio of hydrocarbons, including toluene/benzene and ethylbenzene/benzene (Chu et al., 2021; Parrish et al., 2007). Since the oxidation rates of these hydrocarbons by hydroxyl (OH) radicals span several orders of magnitude, the hydrocarbon ratios change with photochemical aging (Chen et al., 2021). By this definition, the photochemical age determines the degree of photochemical processing associated with OH radicals rather than the physical age of pollutants (Irei et al., 2016).

A few attempts were made to track the age distribution of aerosols using chemical transport models (CTMs) (Han and Zender, 2010; Wagstrom and Pandis, 2009; Wu et al., 2017). CTMs can reproduce the evolution of pollutants in the atmosphere (including emission, transport, deposition, and chemical transformation). Zhang et al. (2019a) introduced a dynamic age-bin updating algorithm in the source-oriented University of California, Davis/California Institute of Technology (UCD/CIT) air quality model to track the age distribution of primary $PM_{2.5}$. In their study, chemical variables in UCD/CIT model were expanded with one more dimension to represent pollutants with different atmospheric ages. More recently, this dynamic age-bin updating algorithm was expanded to include gaseous precursors to determine the age distribution of all primary and secondary inorganic compounds in the Community Multiscale Air Quality (CMAQ) model (Ying et al., 2021). In this study, we further developed an age-resolved UCD/CIT model to track the atmospheric age distribution of various primary and secondary components of $PM_{2.5}$ based on the method used in the CMAQ model. Then we applied the model to investigate the evolution of the concentrations and ages of the major $PM_{2.5}$ components during a typical winter haze episode in eastern China.

## 2. Methods

### 2.1 Description of UCD/CIT Model

The source-oriented UCD/CIT air quality model (Held et al., 2004; Hu et al., 2014; Hu et al., 2015; Kleeman and Cass, 2001; Ying et al., 2007; Ying and Kleeman, 2006) was used in this study to simulate air quality in eastern China. The UCD/CIT model is a 3-dimensional Eulerian regional CTM with detailed chemistry and aerosol mechanisms. Details about the fundamental algorithms used in UCD/CIT model can be found in the above references. Briefly, gas-phase chemistry is modeled by the SAPRC-11 chemical mechanism (Carter and Heo, 2013). Aerosols are represented using a sectional approach with 15 log-spaced size bins encompassing 10 nm–10 μm. Thermodynamic equilibrium for inorganic aerosols is calculated by ISORROPIA (Nenes et al., 1998). Secondary organic aerosol (SOA) treatment is based on the two-product model used in the Community Multiscale Air Quality (CMAQ) model, including a total of 19 semi-volatile or nonvolatile species from seven precursors (Carlton et al., 2010).

In most existing air quality models, particles from diverse emission sources are mixed. However, the UCD/CIT model applies a source-oriented framework in which primary and secondary particles from each source category are tracked separately through the calculation of all major atmospheric processes, such as advection, diffusion, deposition, and gas-particle partitioning. Thus, the source contributions to regional particle concentrations can be evaluated. Zhang et al. (2019a) expanded the source-oriented UCD/CIT model to track the age distribution of elemental carbon (EC) in the atmosphere. In this study, we implemented the atmospheric age distribution modeling framework (Ying et al., 2021) to track the age distribution of both primary and secondary aerosols. The age distribution of SOA was added to the framework, making it possible to have complete age-resolved modeling of secondary aerosols.

The age-resolved UCT/CIT model employs a sectional approach to track particles of different atmospheric ages (*n* age bins). Freshly emitted or formed particles at each model time step are set to the lowest age bin. At a fixed age-bin updating frequency ($\Delta\tau$), particles in the lower

age bin would be moved to the higher age bin successively. Particles in the last age bin represent those older than the highest explicit age. The average age of particles in the *i*th age bin ( $\overline{\tau_i}$ ) is approximately equal to the middle of that period:

$$\overline{\tau_i} = (i - \frac{1}{2})\Delta\tau, \quad i = 1, 2, ..., n \tag{1}$$

The average age of particles can be calculated by

$$\overline{\tau} = \sum \frac{\tau_i C_i}{\sum C_i} \tag{2}$$

## 2.2 Model setup

The age-resolved UCD/CIT model was run from 21 December 2017 to 2 January 2018, with the first 4 days as the spin-up period to minimize the impact of initial conditions. The model domain has a horizontal resolution of 36 km encompassing eastern China and a vertical structure of 16 layers with 10 layers below 1 km. Hourly meteorological inputs were generated by the Weather Research Forecasting (WRF) model version 4.2 with initial and boundary conditions from the 1.0° × 1.0° National Centers for Environmental Prediction Final (NCEP FNL) operational global analyses dataset. More details on the WRF model configuration can be found in Xie et al. (2022a).

Anthropogenic emissions were taken from MEIC (the Multi-resolution Emission Inventory for China) v1.3 with a spatial resolution of 0.25° ×0.25° (Zheng et al., 2018). FINN (Fire INventory from NCAR) v1.5 with 1 km resolution (Wiedinmyer et al., 2011) and MEGAN (Model of Emissions of Gases and Aerosols from Nature) (Guenther et al., 2006) driven by meteorological inputs from WRF were used to provide wildfire and biogenic emissions, respectively. Total particle-phase emissions from the above mentioned sources were transformed into size-resolved emissions based on measured source profiles (Kleeman et al., 2008; Robert et al., 2007a; Robert et al., 2007b). In addition, sea salt and dust emissions were calculated online within the model based on wind speed and land use type, as described in Hu et al. (2015).

A total of 9 age bins were configured to determine the age distribution of particles in this study. The age-bin updating frequency was set to 12 h in our base simulation so that we could explicitly track particle ages up to 96 h. However, $PM_{2.5}$ concentrations can grow explosively during our study period within several hours. Thus, another four simulations with age-bin updating intervals of 1, 3, 6, and 8 h were also conducted to better reflect the age distribution of particles. Results from different simulations were combined by replacing the low time resolution simulations with the corresponding high time resolution results (Ying et al., 2021). The simulated concentrations of $PM_{2.5}$ and its major components from the age-resolved model show good agreement with the original UCD/CIT model (Figure S1), confirming that the dynamic age-bin updating algorithm will not change the concentration prediction. The computational burden of the age-resolved UCD/CIT model with 9 age bins is ~3 times slower than the original model.

### 2.3 Field observations

Hourly meteorological data, including 2-m air temperature (T2), 10-m wind speed (WS) and direction (WD), and 2-m relative humidity (RH) were collected from 171 routine weather stations in eastern China (**Figure 1**) from the Chinese National Meteorological Center (http://data.cma.cn/en). Surface $PM_{2.5}$ observations were acquired from the national air quality monitoring network developed by the China National Environmental Monitoring Center (http://www.cnemc.cn/en). Measurements on aerosol composition were conducted at four main cities in eastern China, including Beijing, Jinan, Nanjing, and Shanghai. Water-soluble inorganic ions (WSIIs, including nitrate ($NO_3^-$), sulfate ($SO_4^{2-}$), and ammonium ($NH_4^+$)) were measured by an online analyzer (MARGA, model ADI 2080, Applikon Analytical B. V. Corp., Netherlands) with a $PM_{2.5}$ cyclone inlet in Jinan, Nanjing, and Shanghai (Shu et al., 2019). In Beijing, the mass concentrations of WSIIs were analyzed by two ion chromatography systems (DIONEX ICS2000 and ICS2500 for cations and anions, respectively) (Tan et al., 2018). Carbonaceous components (organic carbon (OC) and EC) in four cities were analyzed with a carbon analyzer (model RT-4, Sunset Laboratory Inc., USA) based on the thermal-optical

transmittance method (Wang et al., 2016b). More details about the principles and operation of the above instruments can be found in the corresponding references.

The incremental mass ratio (IMR) proposed by Tan et al. (2018) was adopted in this study to determine the aerosol components that drive the particle concentration growth during the haze episode. Briefly, the IMR of a certain component $i$ ($IMR_i$) is calculated as the ratio of the increment of component $i$ ($\Delta C_i$) to the increment of $PM_{2.5}$ ($\Delta PM_{2.5}$) total mass during the $PM_{2.5}$ growth process:

$$IMR_i = \frac{\Delta C_i}{\Delta PM_{2.5}} \times 100 \qquad (3)$$

Thus, the contribution of each chemical composition to the $PM_{2.5}$ increment can be calculated.

## 3. Results

### 3.1. Episode description and model evaluation

**Figure 2** shows a severe regional haze episode over eastern China spanning from 25 December 2017 to 2 January 2018. The time series of $PM_{2.5}$ concentrations in eastern China indicates that $PM_{2.5}$ gradually accumulated in the NCP from 25–28 December 2017 under the condition of low wind speed (~2 m $s^{-1}$) and increasing RH (**Figure S1**), which is identified as the accumulation stage. Severe haze pollution characterized by high $PM_{2.5}$ concentrations (> 150 μg $m^{-3}$) persisted from the night of 28 December to the morning of 30 December, while the peak value of $PM_{2.5}$ reached 191 μg $m^{-3}$ at 10:00 LT 29 December (stabilization stage). On 30 December, a cold front formed in the NCP, where the cold air in front of the Siberian High encountered the warm air from the south (**Figure S2**). As a result, the wind speed increased sharply from 2.5 m $s^{-1}$ to 5.7 m $s^{-1}$ within 6 h, followed by a steep drop in air temperature from 4.3 °C to −7.0 °C (**Figure S1**). Under the influence of strong northwesterly winds, a continuous movement of $PM_{2.5}$ from north to south (i.e., Taiyuan, Linfen, Shijiazhuang, Zhengzhou, Nanjing, and Shanghai) occurred, and the polluted air masses dissipated quickly in the NCP within several hours (dilution stage) (**Figure S3**). Consequently, severe haze pollution formed

rapidly in the YRD during 30–31 December due to regional transport from the NCP, with the peak value of $PM_{2.5}$ concentrations greater than 200 μg $m^{-3}$.

To better explore the characteristics of $PM_{2.5}$ pollution in the YRD, the haze episode was divided into three stages (before, during, and after regional transport) in this study according to $PM_{2.5}$ concentrations and winds (**Figure 2a and 2b**). Before regional transport, $PM_{2.5}$ concentrations in the NCP (> 250 μg $m^{-3}$) were much higher than those in the YRD (~70 μg $m^{-3}$). Low wind speed (~2 m $s^{-1}$) favored the accumulation of air pollutants in the NCP. Meanwhile, southeasterly winds prevailed in the coastal areas of the YRD, bringing less polluted air masses. In the following 1–2 days, eastern China was under the control of strong northwesterly winds (4–5 m $s^{-1}$) due to the cold front, and the heavily polluted air masses gradually moved from north to south (**Figure 2d**). After the cold front passes, high pressure controls the YRD, leading to subsidence and trapping $PM_{2.5}$ in the PBL. Thus, high concentrations of $PM_{2.5}$ occurred in the YRD with low wind speed, especially in Jiangsu and Shanghai (**Figure 2e**).

The UCD/CIT model well reproduces the observed temporal variations of hourly $PM_{2.5}$ concentrations averaged over the NCP and the YRD during this haze episode with a high correlation coefficient (R > 0.85) and a low bias (NMB < 15%) (**Figure 2b**). High $PM_{2.5}$ concentrations (> 150 μg $m^{-3}$) with low wind speed over southern Hebei, Shandong, Henan, northern Jiangsu, and Anhui provinces are well captured by the model (**Figure S4**). The simulated $PM_{2.5}$ compositions ($SO_4^{2-}$, $NO_3^-$, $NH_4^+$, EC, and organic matter (OM)) also agree well with the daily-averaged measurements in Beijing, Jinan, Nanjing, and Shanghai (**Figure S5**), with model performance statistics comparable to those in other studies(Shi et al., 2017; Hu et al., 2016; Zhang et al., 2019b). Detailed model evaluation can be found in the Supporting Information.

### 3.2. Evolution of particle chemical compositions

**Figure 3** shows the observed chemical composition evolution in Beijing, Jinan, Nanjing, and Shanghai. SNA increased rapidly and became the major component of $PM_{2.5}$ during the $PM_{2.5}$

growth process in all four cities, while the mass fraction of EC and OM in $PM_{2.5}$ decreased. In Beijing and Jinan, located in the NCP, the daily averaged SNA concentrations increased from 10 and 22 μg $m^{-3}$ on 25 December to 110 and 157 μg $m^{-3}$ on 29 December, and their mass fraction in $PM_{2.5}$ increased from ~40% to ~75%. During 25–29 December, the NCP region was under the control of a uniform pressure field with low horizontal winds (**Figure S2**), and pollutants gradually accumulated under such stagnant conditions. The observed RH gradually increased, and the maximum exceeded 80% on 28–29 December, which facilitated the chemical formation of secondary aerosols and accelerated the hygroscopic growth of particles (Cheng et al., 2016; Sun et al., 2014; Yang et al., 2015). Process analysis also indicates that chemical formation is the driving process for the growth of SNA in the NCP, with its net production rate ~3 times larger than that of vertical mixing and horizontal advection during the accumulation stage (**Figure 4a, b**). For YRD cities, Nanjing and Shanghai, daily SNA concentrations increased sharply by 3–6 times within two days (30–31 December) and accounted for 78% of the peak $PM_{2.5}$. Horizontal advection played a dominant role during the explosive growth of air pollutants, with a maximum production rate of 8.1 and 2.7 μg $m^{-3}$ $h^{-1}$ for $NO_3^-$ and $SO_4^{2-}$ respectively (**Figure 4c, d**). The chemical process also contributed obviously to $NO_3^-$ and $SO_4^{2-}$ in Nanjing during the regional transport, indicating the continuous local formation in the YRD.

$NO_3^-$ exhibited the highest levels among SNA in all four cities. The peak value of $NO_3^-$ was 49, 57, 80, and 51 μg $m^{-3}$ for Beijing, Jinan, Nanjing, and Shanghai, respectively, contributing to 25–41% of $PM_{2.5}$ mass concentrations. The IMR of $NO_3^-$ (29–33%) was much higher than that of other components in Beijing, Nanjing, and Shanghai (**Figure 3e**), indicating that $NO_3^-$ was the driving component during the $PM_{2.5}$ growth process. In Jinan, the IMR of $SO_4^{2-}$ (26%) was slightly higher than that of $NO_3^-$ (24%). Nevertheless, the mass fraction of $NO_3^-$ in $PM_{2.5}$ during the $PM_{2.5}$ growth process (25–29 December) was 26%, significantly larger than that of $SO_4^{2-}$ (16%) and $NH_4^+$ (15%). The higher fraction of $NO_3^-$ in $PM_{2.5}$ and its dominant contribution to the $PM_{2.5}$ growth process have also been pointed out by recent observation and modeling studies conducted in eastern China during winter haze periods (Shao et al., 2018; Xu

et al., 2019; Xie et al., 2022b).

### 3.3. Evolution of particle age distribution

The age distribution evolutions of the major $PM_{2.5}$ compositions (EC, $SO_4^{2-}$, $NO_3^-$, $NH_4^+$, POA, and SOA) in Beijing and Shanghai are illustrated in **Figures 5 and 6**. High concentrations of EC, POA, and $SO_4^{2-}$ typically occurred at low atmospheric ages, with obvious diurnal variations in both cities. The bimodal distribution of fresh particles was mainly related to the variations in local emissions and the evolution of PBL. Because the shallow PBL at night was not conducive to the diffusion of air pollutants, particles emitted during evening rush hours remained in the atmosphere longer than those emitted in the morning. Several hours after being released, the concentrations of particles dropped rapidly due to the atmospheric dilution process such as advection and deposition. Nevertheless, under stable weather conditions with low wind speeds, the dilution effect was weak, and EC, POA, and $SO_4^{2-}$ particles could accumulate in the atmosphere for a longer time. This can be seen in Beijing from 25–29 December with a gradually increasing mean atmospheric age of EC, POA, and $SO_4^{2-}$. During this time, relatively high concentrations of aged EC, POA, and $SO_4^{2-}$ particles (with atmospheric age > 24 h) together with large contributions from fresh particles (with atmospheric age < 24 h) can also be observed (e.g., 12:00 to 16:00 LT 28 December). On 30 December, aged EC, POA, and $SO_4^{2-}$ particles in Beijing were removed sharply by strong northwesterly wind, leading to a steep decrease (from ~40 h to less than 6 h) in their mean atmospheric age. Subsequently, in Shanghai, the concentration of aged particles increased rapidly during the period from 14:00 LT on 30 December to 02:00 LT on 31 December, indicating the influence of regional transport. As a result, the mean atmospheric age of EC, POA, and $SO_4^{2-}$ increased from 3–6 h to 47–52 h.

Similar to that of EC, POA, and $SO_4^{2-}$, the average age of SOA gradually increased during the accumulation stage in the NCP region, and then decreased sharply on 30 December due to the sweeping effect of the strong northwesterly wind. In YRD, the average age of SOA before the regional transport was larger than that of EC and POA, although its concentrations were

relatively low. During the regional transport, the average age of SOA increased from ~20 h to ~60 h within several hours. Oligomers of anthropogenic SOA (AOLGA), xylene, toluene, long-chain alkanes, and monoterpenes were found to be the most important precursors, contributing over 95% of the total SOA in Beijing, Jinan, Nanjing, and Shanghai (**Figure S7 and S8**). The contribution of AOLGA increased with atmospheric age in all four cities, while the contributions of xylene, toluene, and long-chain alkanes decreased with age. In the NCP cities, the contribution of AOLGA to total SOA concentrations increased from 35% in the accumulation stage to 48% in the stabilization stage. This is because semi-volatile SOA would not immediately form AOLGA after being released into the atmosphere, and the oligomerization reactions take time.

$NO_3^-$ and $NH_4^+$ exhibited different age distributions compared to EC, POA, $SO_4^{2-}$, and SOA. The mean atmospheric age of $NO_3^-$ and $NH_4^+$ did not show an increasing trend and even decreased on some occasions during the accumulation stage in Beijing and Jinan (**Figures S9 and S10**). Such a low atmospheric age was mainly due to the continuous formation of secondary $NH_4NO_3$ particles locally (**Figure S11**), which increased the concentrations of fresh particles and decreased the mean age of $NO_3^-$ and $NH_4^+$. High concentrations of fresh $NO_3^-$ combined with a moderate contribution of aged $NO_3^-$ can also be seen during the explosive growth stage (e.g., 16:00 to 23:00 LT 30 December 2017) in Nanjing and Shanghai, indicating that the continuous local formation with additional help from regional transport together contribute to the high concentrations of $NO_3^-$. The average atmospheric age of $NO_3^-$ in Jinan, Nanjing, and Shanghai decrease significantly with RH and low average ages are often observed with a high concentration of $NO_3^-$ (**Figure S12**). This confirms our speculation that the rapid chemical formation of $NO_3^-$ under high RH conditions (**Figure S1**) leads to a high concentration of fresh $NO_3^-$ and decreases the mean age of $NO_3^-$.

The vertical cross sections of the concentrations and mean ages for EC and $NO_3^-$ along the transport route from Beijing to Shanghai (white solid line in **Figure 1**) are shown in **Figures 7 and 8**, while those for $SO_4^{2-}$ and SOA are illustrated in **Figures S13 and S14**. Before the regional transport (e.g., 16:00 LT 28 December 2017), aged particles mainly accumulated in

the NCP under slow wind speed (~2 m $s^{-1}$). The average age of EC, $SO_4^{2-}$, $NO_3^-$, and SOA was approximately 30–50 h, 50–60 h, 20–45 h, and 50–60 h respectively, larger than that in the YRD (15–30 h for EC, 30–50 h for $SO_4^{2-}$, 8–24 h for $NO_3^-$, and 35–45 h for SOA). An obvious vertical gradient of particle age occurred in the YRD, with lower age near the surface and higher age aloft. Due to the accumulation of aged pollutants in the atmosphere under stable weather conditions, the vertical distribution of particle ages was more uniform in the NCP. When the cold front passed through, polluted air masses carrying aged particles gradually moved from the NCP to the YRD under strong northwest wind (>5 m $s^{-1}$). During this period, high $PM_{2.5}$ concentrations occurred in eastern China and vertically extended up to 1.2 km, forced by the upward motion along the cold front. For EC, $SO_4^{2-}$, and SOA, the average age in the YRD increased significantly within the whole PBL due to the regional transport from the NCP. The maximum age reached 48, 60, and 65 h for EC, $SO_4^{2-}$, and SOA, respectively. For $NO_3^-$, the highest concentration reached 180 μg $m^{-3}$ for regions between Jinan and Nanjing at 08:00 LT on 30 December 2017, which was 2 times the peak value in the NCP before the regional transport (~90 μg $m^{-3}$). Moreover, the average age of $NO_3^-$ was relatively low (6–24 h) for high-concentration $NO_3^-$ particles, indicating the significant contribution from the local chemical formation.

The age distribution of the major $PM_{2.5}$ chemical compositions (EC, $SO_4^{2-}$, $NO_3^-$, $NH_4^+$, and SOA) in Beijing, Jinan, Nanjing, and Shanghai is shown in **Figure 9**. $SO_4^{2-}$ and SOA exhibited larger atmospheric age than the other three species, with a maximum average age of 84 and 81 h, respectively. In the NCP cities, Beijing and Jinan, more aged particles occurred in the stabilization stage. The average age in Beijing was 45, 76, 55, 61, and 71 h for EC, $SO_4^{2-}$, $NO_3^-$, $NH_4^+$, and SOA respectively, higher than that in the accumulation and dilution stages. It is worth noting that a large fraction (50–60%) of fresh $NO_3^-$ and $NH_4^+$ particles with an age of less than 12 h occurred in Jinan during the stabilization stage, indicating a rapid local formation. In the YRD cities, Nanjing and Shanghai, the mass fraction of aged EC, $SO_4^{2-}$, and SOA particles increased significantly during the regional transport, their average ages were even larger than that in the NCP. This is mainly because of the strong northwesterly wind that

brought abundant aged particles from the NCP. $NO_3^-$ and $NH_4^+$ showed smaller atmospheric age than $SO_4^{2-}$ and SOA, with an average age of 20–30 h during the regional transport. Fresh $NO_3^-$ and $NH_4^+$ particles with atmospheric age of less than 24 h account for more than 70% of the total mass.

**Figures 10 and S15** show the size distribution of the major $PM_{2.5}$ chemical compositions in Beijing and Shanghai. Both EC and POA exhibited bimodal distributions, with a fine-mode peak at 0.2–0.4 μm and a coarse-mode peak at 1–4 μm, respectively. SOA, $SO_4^{2-}$, $NO_3^-$, and $NH_4^+$ were mainly concentrated in the fine mode, with a peak at 3–4 μm. The size distribution of particles with different atmospheric ages was quite different. Aged particles were mainly concentrated in a larger size range, especially for SOA, $SO_4^{2-}$, and $NO_3^-$. For example, $SO_4^{2-}$ with a diameter >0.4 μm in both Beijing and Shanghai showed an atmospheric age of >96 h. When the accumulation stage evolved into the stabilization stage, the size of SOA, $SO_4^{2-}$, and $NO_3^-$ slightly increased in Beijing, while that of EC and POA remained almost unchanged. In Shanghai, $NO_3^-$ and $NH_4^+$ were mainly concentrated in the size range of 0.1–0.3 μm before the regional transport. Their dominant size increased to 0.3–0.7 μm during the regional transport.

## 4. Discussion

Our results indicate that the atmospheric age of EC, POA, $SO_4^{2-}$ and SOA increased gradually during the accumulation stage in the NCP due to air stagnation. The regional transport from the NCP to the YRD brought in high concentrations of aged primary particles, such as EC and POA. As a result, the simulated average atmospheric age of EC was ~40 h during the regional transport, which was much higher than the 'experimental' aging time scale to achieve complete morphology modification and absorption enhancement of BC in Beijing (4.6 h) and Houston (18 h) (Peng et al., 2016). It could be speculated that the aged EC or POA particles are coated continuously by the newly formed fresh SNA particles along the transport route, which could further enhance the light absorption of particles (Bond et al., 2013). Using transmission electron microscopy (TEM), Zhang et al. (2021) observed abundant spherical primary OM particles coated with secondary aerosols in the YRD during the regional transport, which is consistent with our findings. Previous studies have confirmed the crucial role of aerosol-PBL

interaction in altering the vertical structure of PBL and the formation and accumulation of haze in eastern China (Huang et al., 2020; Li et al., 2017). Thus, the potential absorption enhancement of aged black or brown carbon particles during the regional transport could amplify the aerosol-PBL interactions and further exacerbate air pollution.

Another interesting finding is that the atmospheric age of $NO_3^-$ remained unchanged or even slightly decreased during the regional transport from NCP to YRD, contrary to the age evolution of $SO_4^{2-}$. This indicates that $SO_4^{2-}$ is mainly formed upwind and then transported to YRD, while there is a large fraction of $NO_3^-$ is formed locally in YRD. $SO_4^{2-}$ concentrations have been dramatically reduced during the last decade due to desulfurization devices vigorously promoted in coal-fired facilities, and $NO_3^-$ has become the dominant inorganic component of $PM_{2.5}$ in most regions of Eastern China (Sun et al., 2022). Our previous study for January 2013 suggested that $NO_x$ emissions from local sources and adjacent Jiangsu province contributed to nearly 30% of $NO_3^-$ in Shanghai, respectively (Xie et al., 2021). Therefore, $NO_3^-$ reduction can be achieved by cooperative emission controls within the YRD region. Surely, emission reduction actions should be taken a few days in advance to mitigate severe haze pollution under unfavorable weather conditions.

This study is subject to a few limitations. The UCD/CIT model includes $SO_4^{2-}$ formation mechanisms through the gas phase oxidation of $SO_2$ by OH radicals and the in-cloud aqueous oxidation. However, recent field observations indicated a large contribution from other pathways during winter haze events in China, such as manganese-catalyzed oxidation on aerosol surfaces (Wang et al., 2021b), and aqueous oxidation of $SO_2$ by $NO_2$ (Cheng et al., 2016; Wang et al., 2016a). The missing mechanism in the current model leads to a substantial underestimation of $SO_4^{2-}$ (42.3%), which will further affect the age distribution of $SO_4^{2-}$ particles. Since SOA is universally underestimated in current CTMs (Hu et al., 2017), uncertainties may also occur with SOA. Additionally, the discretization of atmospheric age in our model can lead to some uncertainties, especially for the calculation of average atmospheric age (Xie et al., 2022a). Thus, in this study, we run five different simulations with $\Delta\tau$ of 1, 3, 6,

8, and 12 h respectively, and combined the results to minimize the effect of discrete age representation.

## 5. Conclusions

In this study, the age-resolved UCD/CIT model was used to investigate the age distribution of $PM_{2.5}$ during a severe regional haze episode in eastern China in December 2017. Comparison with surface observation shows that the model reasonably captured the spatiotemporal variations of $PM_{2.5}$ and its major chemical compositions. Our results indicate that at the beginning stage of the haze event (25–29 December 2017), the stagnant weather conditions characterized by weak surface wind and high RH facilitated the accumulation and secondary formation of air pollutants, leading to increased $PM_{2.5}$ concentrations in the NCP region. $NO_3^-$ was found to be the dominant chemical composition during this haze episode, contributing to ~30% of $PM_{2.5}$ concentration growth in Beijing, Jinan, Nanjing, and Shanghai. Both the concentration and atmospheric age of EC, POA, $SO_4^{2-}$ and SOA increased gradually during the accumulation stage in the NCP due to weakened atmospheric diffusion capacity, while the atmospheric age of $NO_3^-$ and $NH_4^+$ remained unchanged because of continuous local formation. During the regional transport stage (30 December), a cold front moved from north to south, bringing aged particles from the NCP to the YRD region and increasing $PM_{2.5}$ concentrations rapidly within hours. Accordingly, the average atmospheric age of EC, POA, $SO_4^{2-}$ and SOA particles in the YRD increased from 5–20 h to 50–60 h. In contrast, continuous local chemical formation resulted in an unexpected decrease in the atmospheric age of $NO_3^-$ and $NH_4^+$ in the YRD, although the concentrations of aged particles with old atmospheric age increased due to regional transport. The age information provided in this study enhances our understanding of the formation mechanism of haze events and helps design cost-effective control strategies for different $PM_{2.5}$ components.

## Code and data availability

Hourly $PM_{2.5}$ data used in this study is freely available through the website of http://106.37.208.233:20035/ (last accessed on January 6, 2023). Meteorological observations

used in this study are available from http://data.cma.cn/en (last accessed on January 6, 2023). The UCD/CIT model outputs are currently available upon request.

**Author contributions**

XX and JH designed research. XX, JH, MQ, HZ and QY contributed to model development, simulations, and data processing. SG, MH, DJ, HW, SL, CH, CL provided the observation data. HL and YZ contributed to result discussion. XX prepared the manuscript and all coauthors helped improve the manuscript.

**Competing interests**

The authors declare that they have no conflict of interest.

**Acknowledgments**

This work was supported by the National Key R&D Program of China (2019YFA0606802), the National Natural Science Foundation of China (41975162, 42277095, 42021004), and the Jiangsu Environmental Protection Research Project (2016015).

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

## Figures

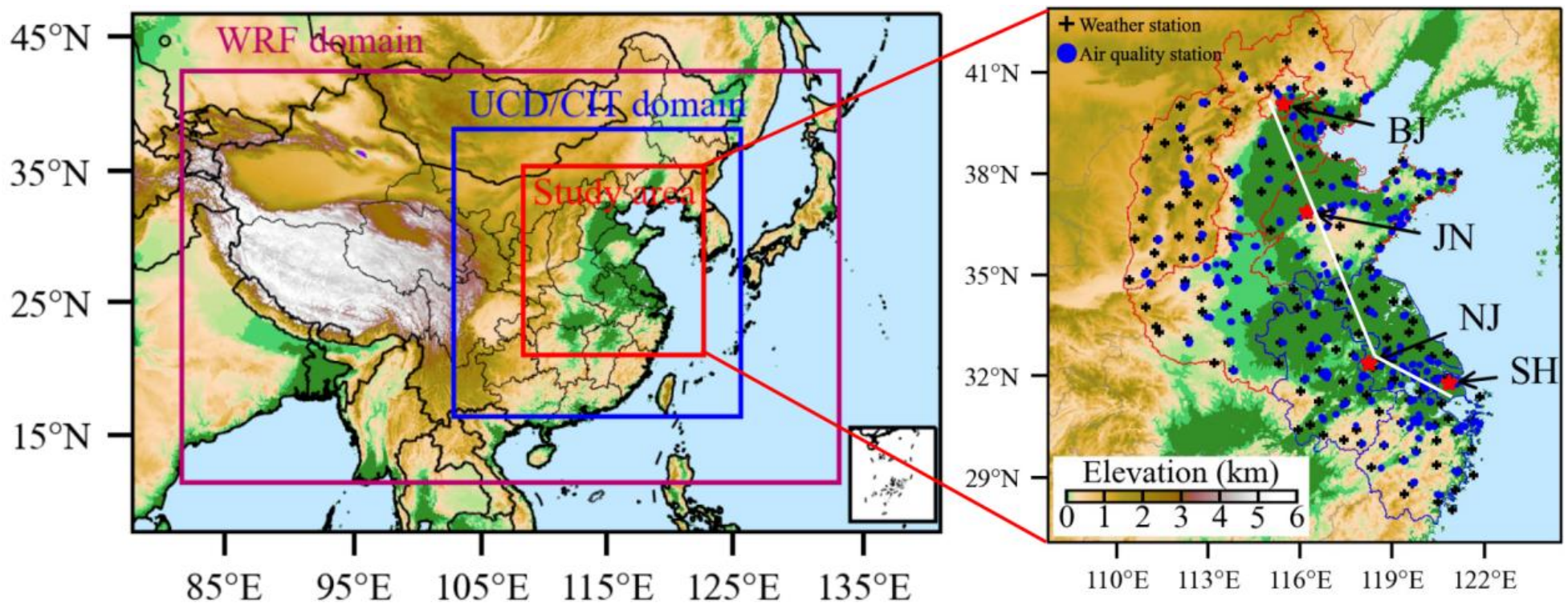

685 **Figure 1.** Modeling domains and the locations of the observation stations. Black crosses represent the weather stations and the blue dots represent the air quality stations. Four main cities (BJ: Beijing, JN: Jinan, NJ: Nanjing, SH: Shanghai) in eastern China are also marked by red stars.

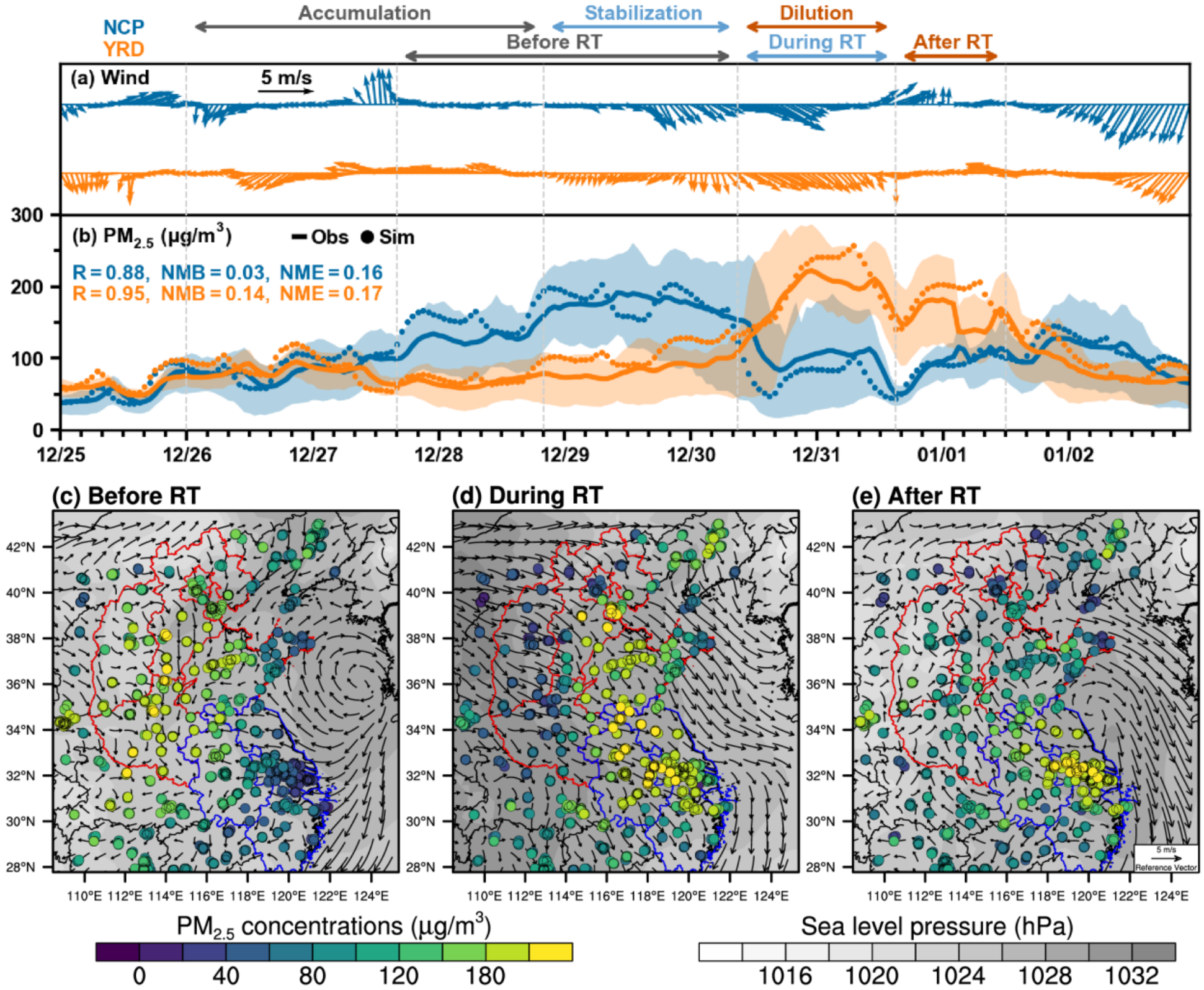

690 **Figure 2**. Time series and spatial distributions of wind fields and $PM_{2.5}$ during this haze episode. **(a, b)** Observed wind and $PM_{2.5}$ concentrations in NCP and YRD. Shaded areas represent the 25th–75th percentile range of observation. Solid dots mark the simulated $PM_{2.5}$ concentrations. **(c–e)** Observed $PM_{2.5}$ concentrations, WRF-simulated wind fields, and sea level pressure before, during, and after regional transport (RT).

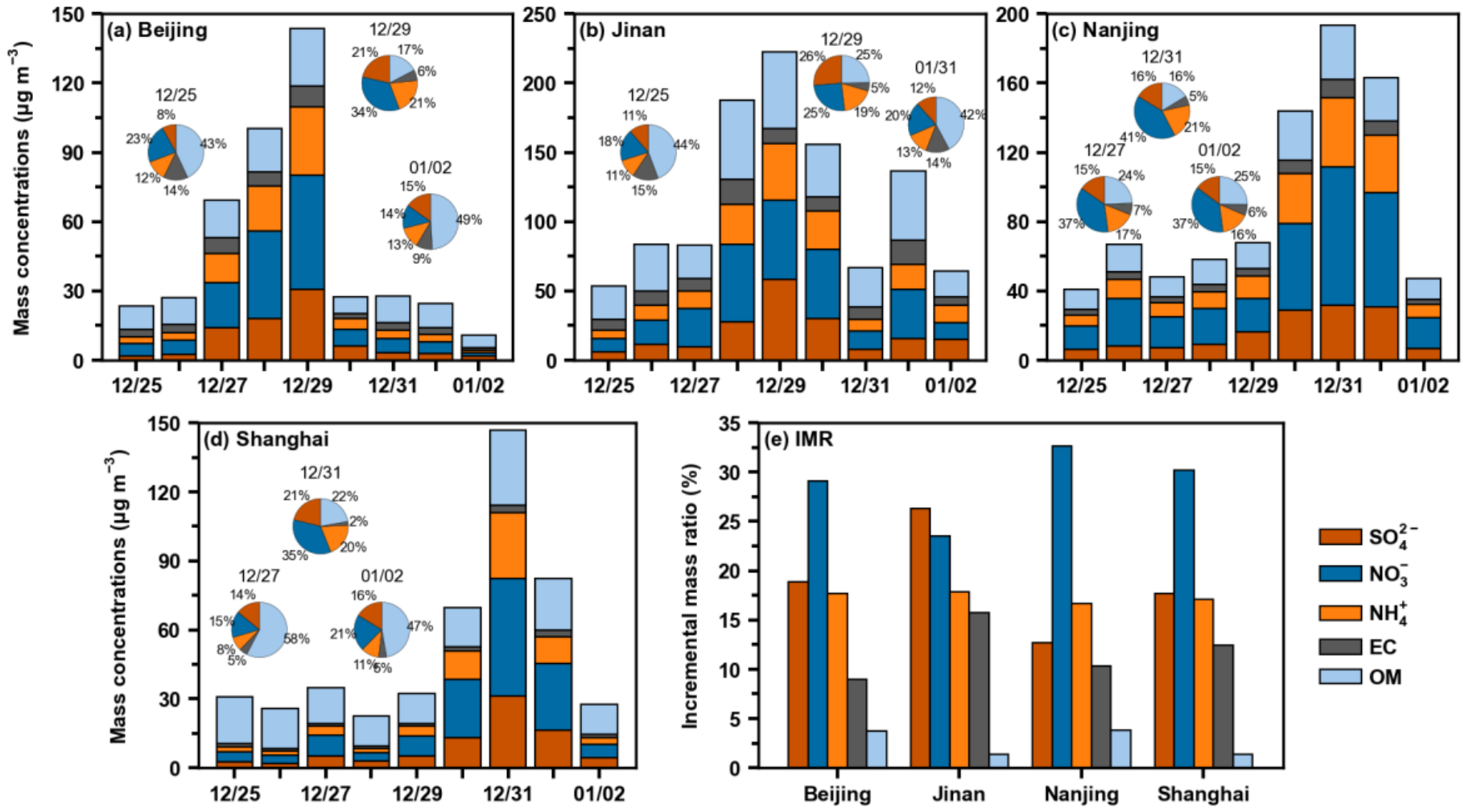

**Figure 3**. Mass concentrations and fractions **(a–d)** and incremental mass ratio (e) of the major $PM_{2.5}$ chemical compositions ($SO_4^{2-}$, $NO_3^-$, $NH_4^+$, EC, and OM) in Beijing, Jinan, Nanjing, and Shanghai.

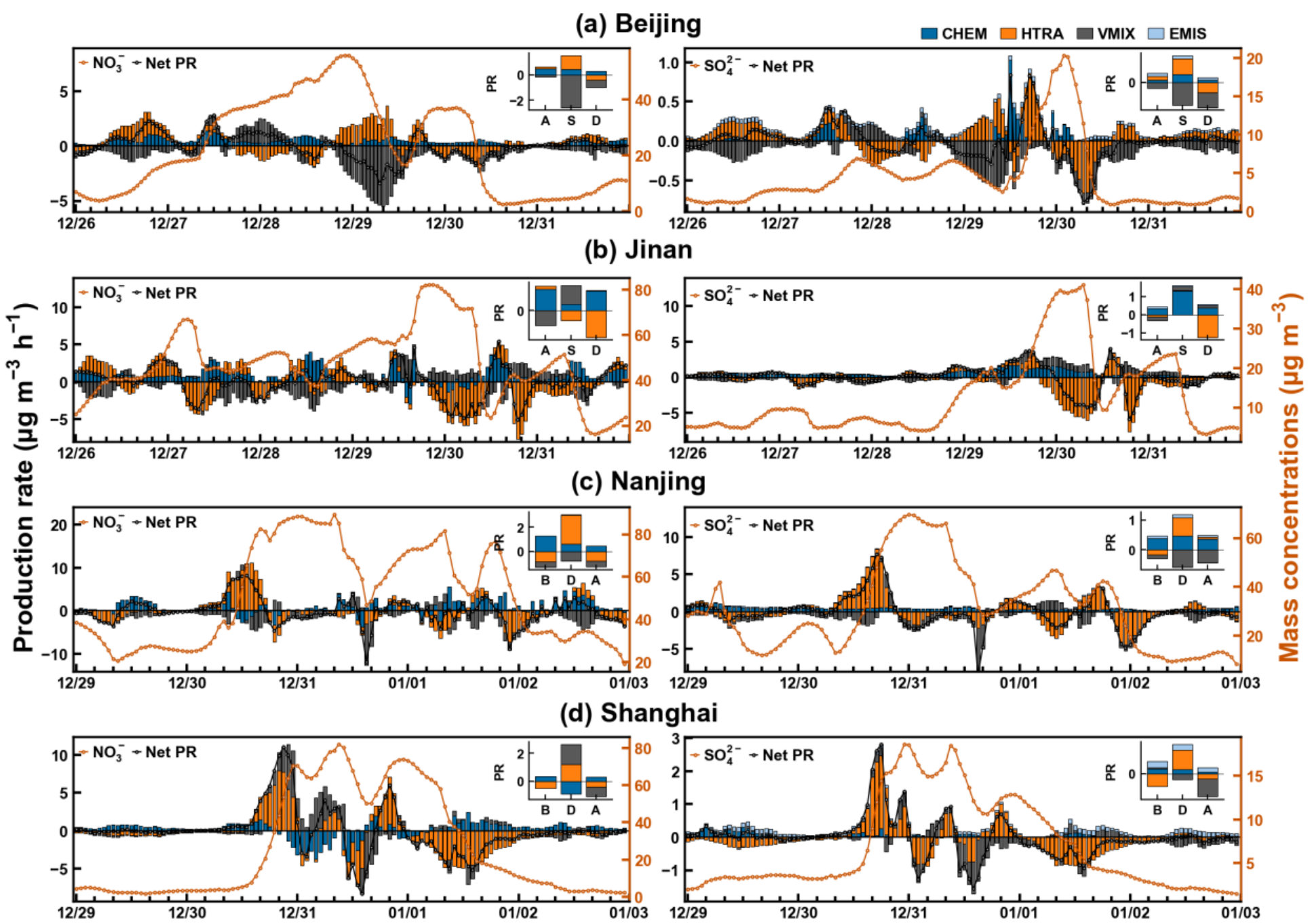

**Figure 4**. The contributions of physical/chemical processes (CHEM: gas-phase, aerosol, and cloud chemistry; VMIX: vertical mixing and dry deposition; HTRA: horizontal advection; EMIS: emission) to $NO_3^-$ and $SO_4^{2-}$ concentrations within the first 10 layers in Beijing, Jinan, Nanjing, and Shanghai. The stacked bar graphs inserted in each panel represent the total contribution during the corresponding period. A, S, and D indicate the accumulation, stabilization, and dilution stage in Beijing and Jinan; and B, D, and A represent the period before, during, and after regional transport in Nanjing and Shanghai.

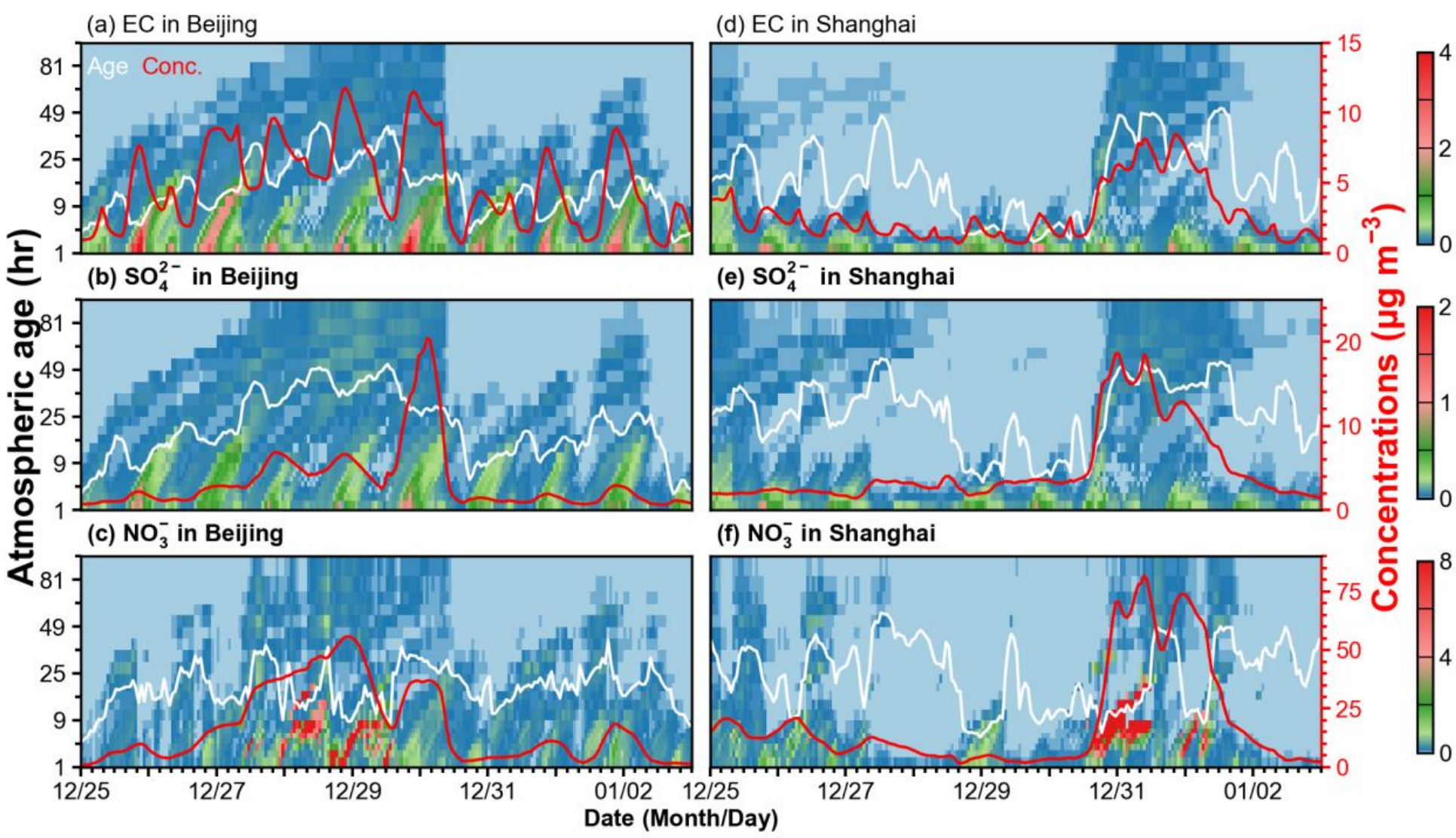

**Figure 5**. Hourly atmospheric age distribution of EC, $SO_4^{2-}$, and $NO_3^-$ in Beijing and Shanghai during this haze episode. White lines represent the average atmospheric age, and red lines (right y-axis) indicate total mass concentrations. The results were combined from simulations with age-bin updating intervals of 1, 3, 6, 8, and 12 h.

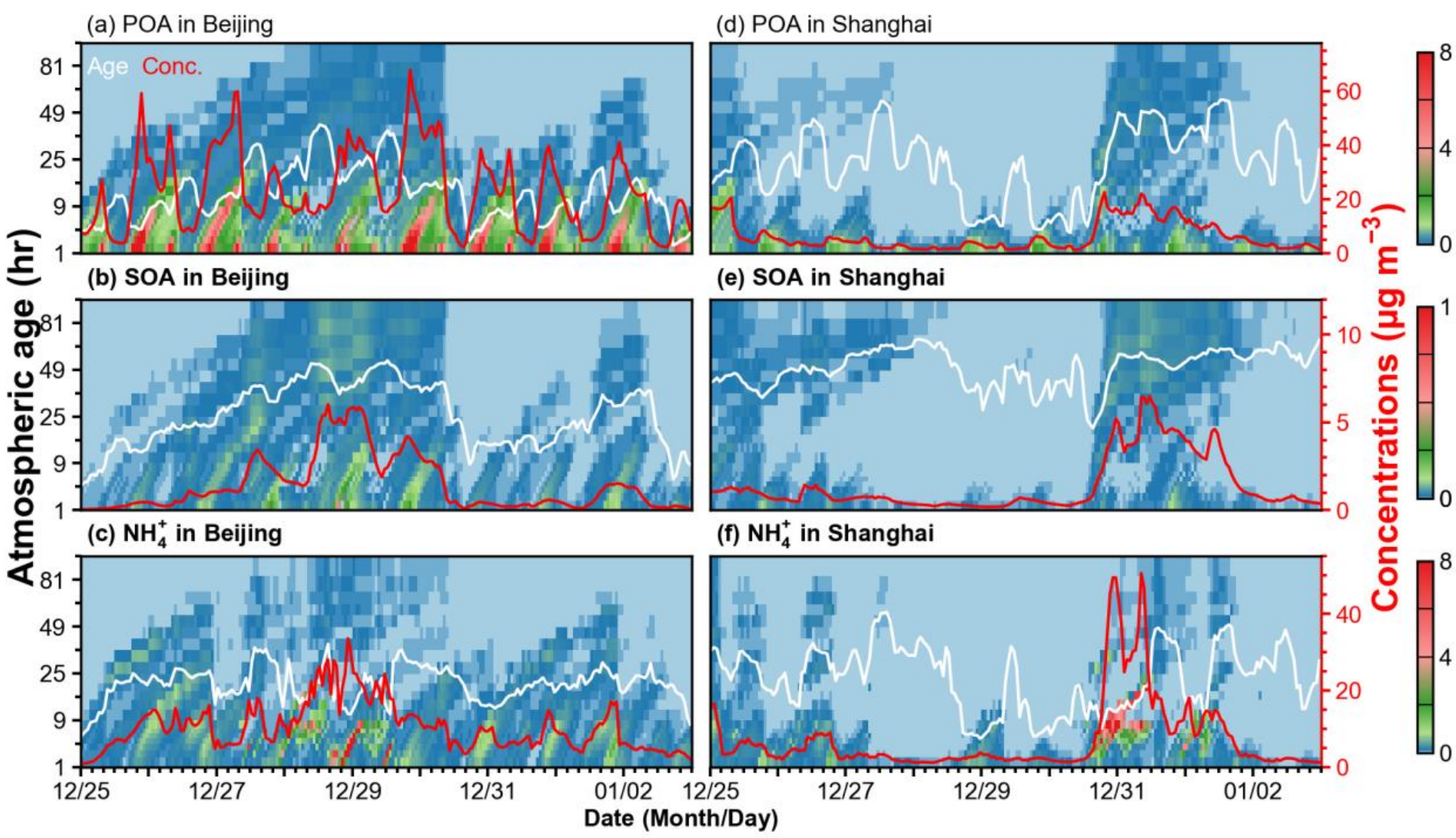

715

**Figure 6.** Same as **Figure 5** but for POA, SOA, and $NH_4^+$.

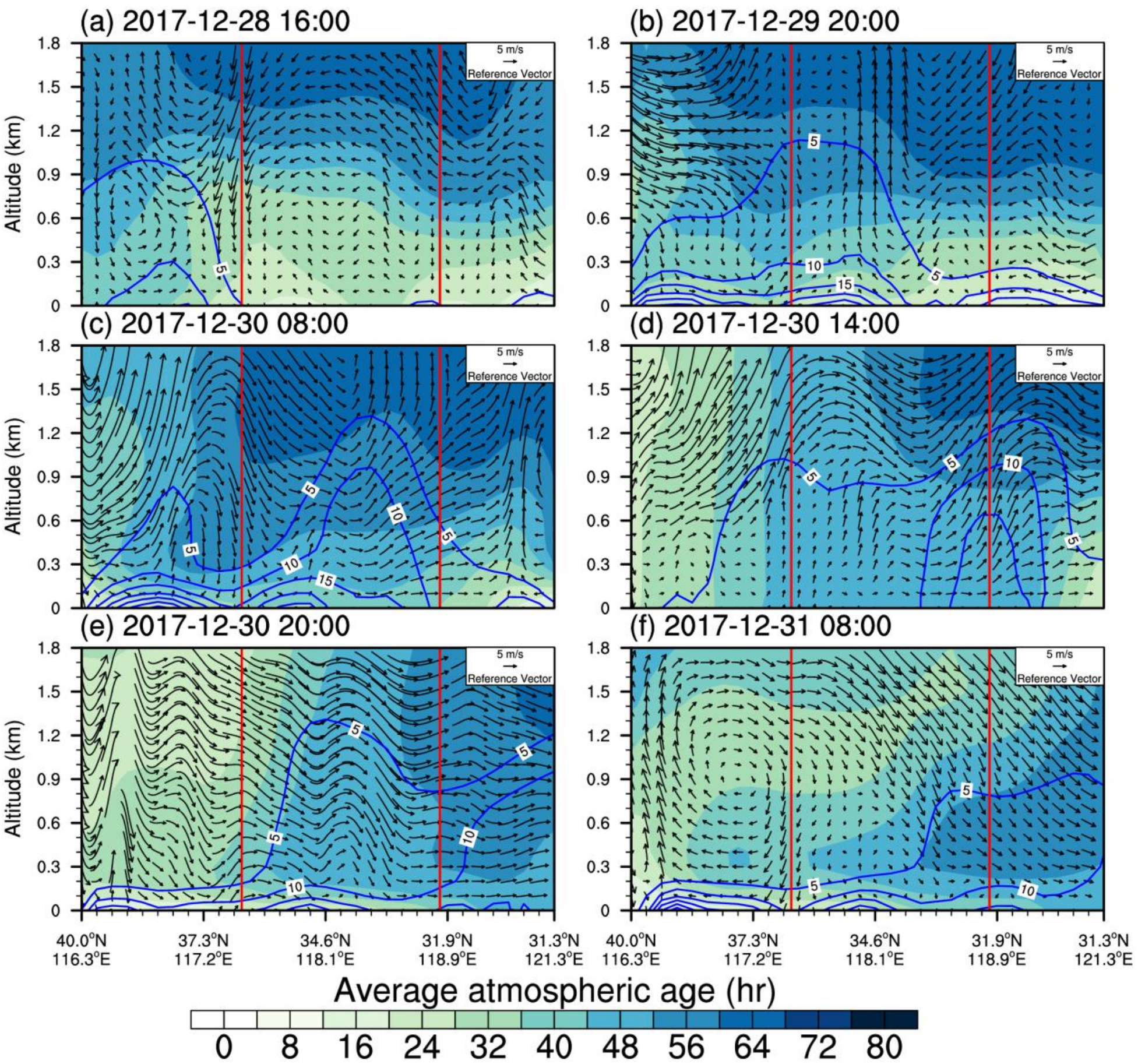

**Figure 7**. Vertical cross section of the average atmospheric age (color contours; h) and concentrations (blue solid lines; μg m$^{-3}$) of EC along the transport route from Beijing to Shanghai (white solid line in **Figure 1**) at **(a)** 16:00 LT 28 December, **(b)** 20:00 LT 29 December, **(c–e)** 08:00, 14:00, 20:00 LT 30 December, and **(f)** 08:00 LT 31 December 2017. Note that the vertical wind speed was multiplied by 500 for the illustration of vertical circulations. The location of Jinan and Nanjing are marked as red solid lines.

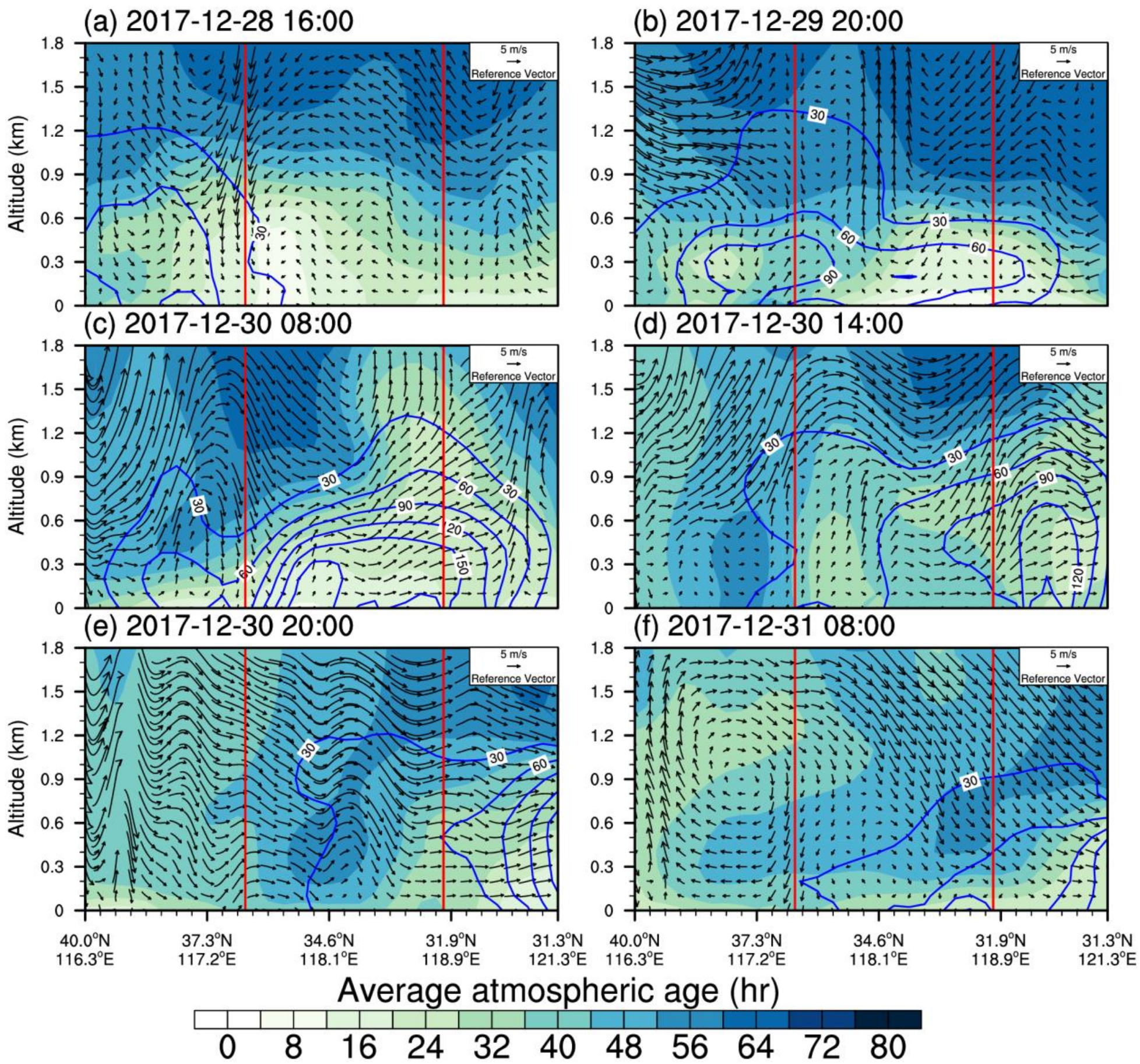

725 **Figure 8.** Same as **Figure 7** but for $NO_3^-$.

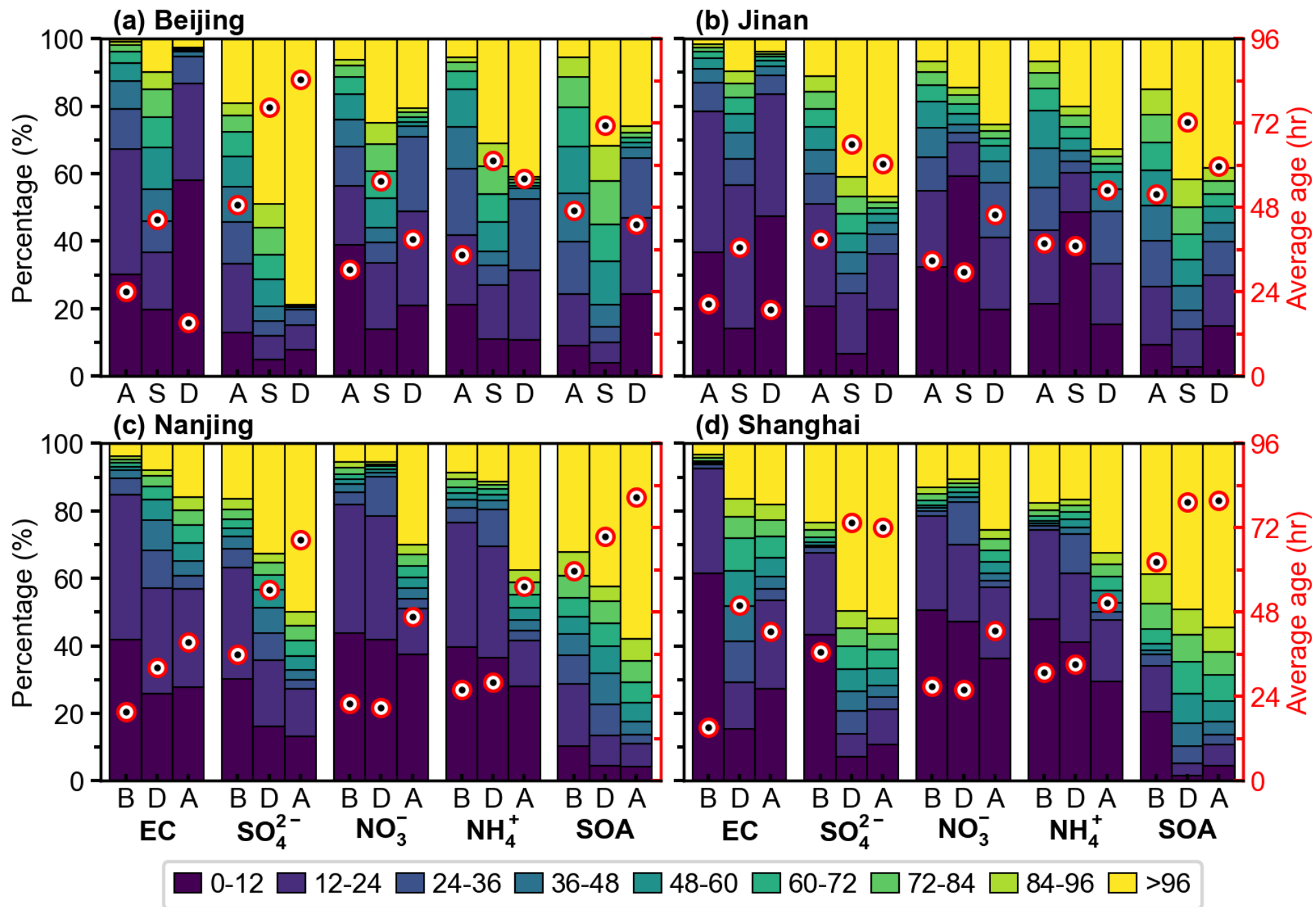

**Figure 9**. The mass fractional contributions of different age bins to EC, $SO_4^{2-}$, $NO_3^-$, $NH_4^+$, and SOA in Beijing, Jinan, Nanjing, and Shanghai. The red circle with a black dot indicates the average atmospheric age (in hours, right y-axis). A, S, and D indicate the accumulation, stabilization, and dilution stage in Beijing and Jinan; B, D, and A represent the period before, during, and after regional transport in Nanjing and Shanghai.

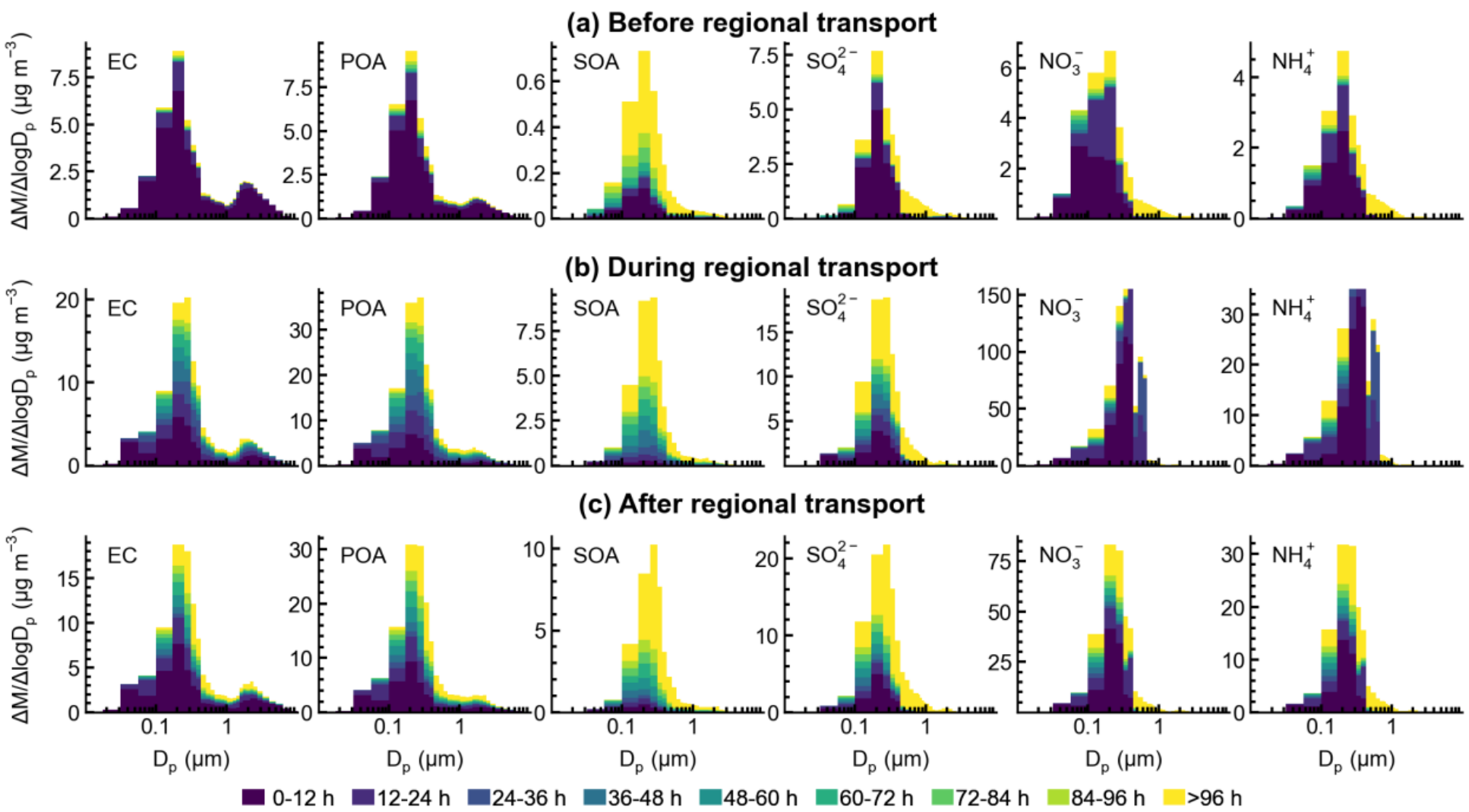

**Figure 10**. The size distribution of EC, POA, SOA, $SO_4^{2-}$, $NO_3^-$, and $NH_4^+$ in Shanghai (a) before, (b) during, and (c) after regional transport.