# Peer review of "Evolution of atmospheric age of particles and its implications for the formation of a severe haze event in eastern China"

_Atmospheric Chemistry and Physics, 2023_

## Author Response (AR1)

Dear Editor and Reviewers,

Thank you for the comments to help improve the quality of the paper. We have revised the manuscript to address your comments and a detailed response to each comment is provided in this file. The comments are in regular font and the responses are in red.

Evolution of atmospheric age of particles and its implications for the formation of a severe haze event in eastern China
Manuscript #: acp-2023-11

**RC1, Reviewer #1:**

This paper presents a simulation-based particle age evolution during a major haze event in China. Although the method is not completely new, the application itself provides some interesting dimensions of the evolution of haze events and demonstrates how particle age information could be used to tackle atmospheric chemistry and physics problems associated with aerosol dynamics. However, there are some concerns regarding the current version of the manuscript. One major issue is the lack of model validation of the predicted particle age, which makes it difficult to determine whether the age numbers should be interpreted as quantitative or qualitative. I would recommend reconsidering this manuscript after the following comments are addressed.

**Response:** Thanks for your valuable feedbacks and insights which have undoubtedly contributed to enhancement of our manuscript. We have made response to each of your comments. Specially regarding your concerns about model validation on the particle age, we absolutely agree that model validation is a crucial step in ensuring the credibility and reliability of the results. We do recognize the importance of comparing our model predictions with real-world measurements to evaluate the accuracy of predicted results. However, currently there are no techniques to measure particle age quantitatively. Therefore, no observational age data is available to directly evaluate the predicted age results. We tried great efforts to provide confidence in interpreting the particle age results by evaluating the predicted results on all available observation data on meteorological parameters, concentrations of $PM_{2.5}$ and itr major chemical compositions over the study period at multiple sites. Since the evolution of particle chemical compositions over time is related to the ages, we assume that the validated chemical composition results can also provide confidence in the predicted age information. We acknowledge this is 'indirect' validation, and have added a short discussion in the revised manuscript (lines 232-238).

Comments:
1. Line 95: this description is confusing. Do you mean you implemented the CMAQ version method that expanded from UCD/CIT back to UCD/CIT?

**Response:** Thanks for the comments. The dynamic age-bin updating method used in our study is based on that previously used in the UCD/CIT model proposed by Zhang et al. (2019). This method has also been used in the CMAQ model (Ying et al., 2021). The difference is that, in CMAQ, the atmospheric age of secondary aerosols is calculated since the emission of precursors, while in UCD/CIT, the atmospheric age of secondary aerosols is calculated since the time they are formed in the atmosphere. To clarify, we have modified this description in the revision.

**Changes in manuscript: (lines 86-100)** Zhang et al. (2019a) introduced a dynamic age-bin updating algorithm in the source-oriented University of California, Davis/California Institute of Technology (UCD/CIT) air quality model to track the age distribution of primary $PM_{2.5}$. In their study, chemical variables in the UCD/CIT model were expanded by adding an additional dimension to represent pollutants with varying atmospheric ages, and the evolution of particle concentrations between different age bins was dynamically updated at a fixed frequency. The dynamic age-bin updating algorithm can be represented in Equation 1.

$$\begin{cases} C^{t+1} = C^t, & t = 1,2,\dots,n \\ C^n = C^n + C^{n-1} \end{cases} \tag{1}$$

Where $t$ is the age bin index, $n$ is the total number of age bins. More recently, this dynamic age-bin updating algorithm was used to determine the age distribution of primary and secondary inorganic compounds in the Community Multiscale Air Quality (CMAQ) model (Ying et al., 2021). In this study, we further developed the age-resolved UCD/CIT model to track the atmospheric age distribution of various primary and secondary components of $PM_{2.5}$ based on the method used by Zhang et al. (2019). Different from that of Ying et al. (2021), the atmospheric age of secondary aerosols is calculated since they are formed in the atmosphere.

2. Line 113: Since primary and secondary particles are tracked separately, does it mean there are no interactions simulated between the primary and secondary particles? Such as coagulation.

**Response:** Thanks for the comments. The interactions between primary and secondary particles, such as condensation and coagulation, are included in the UCD/CIT model. The age distribution of particles defined in this study refers to the mass concentration of particles in different age bins. For the condensation process, particles formed from the condensation of gas phase species would be assigned to the lowest age bins, changing the age distribution of particles. But for the coagulation process, the age distribution of particles would not change.

3. Line 120: CMAQ does not use a sectional method, what is the major challenge or novelty associated with the implementation of adopting the CMAQ method to UCD/CIT?

**Response:** As mentioned in the Response to the comment #1, the age distribution modeling framework is based on Zhang et al. (2019), who also used the UCD/CIT model. The major challenge to simulating the age distribution of particles is how to dynamically update the concentrations between different age bins. To achieve this, a dynamic age-bin updating algorithm (please see Equation 1) is introduced into the UCD/UIT model (Zhang et al., 2019), which has also been used in CMAQ to track the age distribution of primary and secondary inorganic aerosols (Ying et al., 2021). We have removed the citation "(Ying et al., 2020)" in this sentence to avoid misunderstanding. Please see line 123 in the revision.

4. Line 122: How many age bins are used? Is there a limited age range? Does the model allow particles of different sizes and ages to interact with each other? If that is true, how that works?

**Response:** A total of 9 age bins were used in this study, and the age bin updating frequency is set to 12 h. So the explicitly tracked particle age is 96 h. For particles with age > 96 h, their concentrations were assigned to the last age bin. For ages within 96 h, we also conducted another

four simulations with age-bin updating frequency of 1, 3, 6, and 8 h to obtain more detailed age distrbutions. Particles with different sizes and ages can interact with each other through the coagulation process. The coagulation process is expected to change the particle number and size distribution but keep the particle mass concentration constant. Since the particle age distribution defined in this study refers to the mass concentration of particles in different age bins, the age distribution of particles will not change before and after the coagulation process.

5.  Line 132: This is a short period, does it imply that this model is not designed for long-term simulation?

**Response:** Thanks for the comments. The focus of this study is to track the age evolution of particles during haze events and use such information to help understand the formation mechanism of $PM_{2.5}$ pollution. Thus, we selected a typical severe haze episode in eastern China. Note that the age distribution modeling framework used in this study is based on a dynamic age-bin updating algorithm (Equation 1), which can dynamically update the concentrations of particles between different age bins and can be applied for long-term simulations. As mentioned in the Response to the comment #4, the explicitly tracked particle age is 96 h in this study, and the last age bin is used to represent particles older than the highest explicit age (96 h). For long-term simulation, the total number of age bins would not increase, so this dynamic age bin updating method can also be ued for long-term simulation.

6.  Line 135: Figure 1 should be referenced here.

**Response:** Thanks for the suggestion. Added.

7.  Line 154: I am not sure if I understand the age bin updating frequency design strategy here. What is the point to make simulations with different updating intervals? Isn't it obvious that the simulation with updating intervals of 1 h provides the highest age resolution? And how the combination of simulation results works? "By replacing the low time resolution simulations with corresponding high time resolution results", won't that mean everything is replaced by the 1-hour result? In that case, why bother to run simulations of larger intervals? And how do those simulations address uncertainty problems as mentioned in the discussion section?

**Response:** Thanks for the comments. In this study, we use a sectional method to represent particle age, and the total age bins are fixed (9 age bins) in the model. Thus, this method can only represent a limited range of particle age. For example, the simulation with an updating interval of 12 h can explicitly represent particles within 96 h (12×8), and particles older than 96 h are assigned in the last age bin. Note that with an updating interval of 12 h, particles with ages of 0-12 h (12-24h, 24-36h, etc.) are assigned to the same age bin. However, in urban areas, some particles may have been emitted or formed just several hours ago due to intensive anthropogenic emissions. To get relatively detailed age information, we ran several simulations with updating intervals of 1, 3, 6, 8, and 12 h, and then combined them together. As illustrated in table R1-1, we use the corresponding high time resolution simulation when they are available (marked as shaded cells in the table). Although the 1-hour simulation can provide the highest age resolution, the explicitly represented particle age is only 8 hours. Thus, it's not suitable to study the age distribution of particles older than 8 hours.

**Table R1-1.** Average atmospheric age represented by each age bin for simulations with different age bin updating intervals. The grey-shaded cells indicate the results used in the final analysis.

| Age bins | Age bin updating interval (h) | | | | |
|---|---|---|---|---|---|
| | 1 | 3 | 6 | 8 | 12 |
| 0 | 0.5 | 1.5 | 3 | 4 | 6 |
| 1 | 1.5 | 4.5 | 9 | 12 | 18 |
| 2 | 2.5 | 7.5 | 15 | 20 | 30 |
| 3 | 3.5 | 10.5 | 21 | 28 | 42 |
| 4 | 4.5 | 13.5 | 27 | 36 | 54 |
| 5 | 5.5 | 16.5 | 33 | 44 | 66 |
| 6 | 6.5 | 19.5 | 39 | 52 | 78 |
| 7 | 7.5 | 22.5 | 45 | 60 | 90 |
| 8 | 8.5 | 25.5 | 51 | 68 | 102 |

8. Line 160, It doesn't sound right for a "burden" to be "~3 times slower", I would say "the burden is ~ times higher".

**Response:** Thanks for the suggestion. Corrected.

9. Line 180: Does it implies that IMR can be used as a marker of particle age? Is IMR from observation compared with modeled results? Is there a way to check or validated the simulated particle age hours with observation?

**Response:** Thanks for the comments. IMR (incremental mass ratio) is not a marker of particle age. IMR is calculated as the ratio of the increment of a certain chemical component to the increment of $PM_{2.5}$ during the growth stage of the haze episode, which reflects the contribution of an individual chemical component to the growth of $PM_{2.5}$ concentrations. We used IMR to determine which chemical component dominates $PM_{2.5}$ growth. Following the reviewer's suggestion, we've added a new Figure S7 (also shown as Figure R1-1 below) in the Supporting Information to compare the observed and simulated IMR. The simulated IMR agreed well with that of observation, indicating that the model well reproduced the variations in $PM_{2.5}$ chemical components during the haze episode. The particle age defined in this study reflects the instantaneous state of a particle, which is hard to observe or measure. Thus, there are no available observations to directly validate the simulated particle age hours.

[Figure]

**Figure R1-1.** Comparison of the observed (a) and simulated (b) incremental mass ratio of the major PM$_{2.5}$ chemical compositions (SO$_4^{2-}$, NO$_3^-$, NH$_4^+$, EC, and OM) in Beijing, Jinan, Nanjing, and Shanghai.

10. Figure 2, the wind direction (the north I assume) is not marked in the upper panel (a). Also, NCP and YRD are not explained in the map. I am assuming that the red boundary is NCP and the blue boundary is YRD.

**Response:** Thanks for the suggestion. Yes. NCP and YRD refer to the red and blue boundaries in the map, respectively. We've added the explanations in the first paragraph of Section 3.1 (see lines 196-197 and 209) and the captions of Figure 2 (shown as Figure R1-2 below) in the revision. Also, wind direction was added in Figure 2a.

[Figure]

**Figure R1-2**. Time series and spatial distributions of wind fields and PM$_{2.5}$ during this haze episode. **(a, b)** Observed wind and PM$_{2.5}$ concentrations in NCP (red lines in c-e) and YRD (blue lines in c-e). Shaded areas represent the 25th–75th percentile range of observation. Solid dots mark the simulated PM$_{2.5}$ concentrations. **(c–e)** Observed PM$_{2.5}$ concentrations, WRF-simulated wind fields, and sea level pressure before, during, and after regional transport (RT).

11. Line 218: Is the statistic based on hourly results or daily?

**Response:** It is based on the hourly results.

12. Line 228: What is "SNA"?

**Response:** SNA refers to sulfate, nitrate, and ammonium. We've added a description in line 241 in the revision.

13. Line 250: What is the reason for the whole NO3- discussion here? Is it justified why only NO3- is investigated in Figure 4? Some clarification is needed.

**Response:** Thanks for the comments. In section 3.2, we attempt to figure out which chemical component dominates the rapid growth of $PM_{2.5}$ and its underlying driving processes. We found that $NO_3^-$ exhibited the highest concentrations during the $PM_{2.5}$ growth stage in all four cities, and the incremental mass ratio (IMR) of $NO_3^-$ (29–33%) is much higher than that of other components in Beijing, Nanjing, and Shanghai. This indicated that $NO_3^-$ was the driving component during the $PM_{2.5}$ growth stage. Figure 4 is used to quantify the contribution of individual atmospheric processes to $NO_3^-$ and $SO_4^{2-}$ concentrations in the four cities. Since secondary inorganic aerosols increased rapidly during the $PM_{2.5}$ growth stage, both the contributions to $NO_3^-$ and $SO_4^{2-}$ are investigated in Figure 4. We do agree that leaving a whole discussion of $NO_3^-$ in this paragraph is not justified. Thus, we moved the discussion of $NO_3^-$ to the first paragraph in this section and rearranged the structure of this section according to the reviewer's suggestion. Please see Section 3.2 in the revision.

14. Line 260-270, I think the time scale used in Figures 5 and 6 is not very helpful for diurnal observation, I would recommend a 24-hour scale averaged diurnal concentration curve as additional or supporting information. It is difficult to identify the "12:00 to 16:00 LT 28 December" on those figures. Also, when referring to "Beijing" and "Shanghai", do you mean one location in the city or the average information of the whole city domain?

**Response:** Thanks for the comments. We've added a new Figure S8 in the supporting information (also shown as Figure R1-3 below) to show the diurnal variation of the mean atmospheric age and mass concentrations of the major $PM_{2.5}$ chemical compositions. The results indicate that the simulated particle age shows obvious diurnal variations. For most cities, the atmospheric age decreased in the early morning before sunrise. This is because of the increased traffic emissions during local rush hour and the weakened vertical dispersion due to low boundary layer height. The mean atmospheric age increased during the daytime because freshly emitted particles were mixed to a more developed boundary layer. In the late afternoon, the mean atmospheric age decreased again. The increased traffic and residential emissions during evening rush hour and decreased mixing layer height both contribute to the reduction in the mean atmospheric age. The corrending discussion has been added in the revision (please see lines 277-285).

According to the reviewer's suggestion, we've added black lines in Figures 5 and 6 to indicate the period "12:00 to 16:00 LT 28 December" (modified to 12:00 to 18:00 LT 28 December in the revision). Please see Figure R1-4 below. In Figures 5 and 6, "Beijing" and "Shanghai" refer to a specific location in the city, which is based on their geometric center. As a comparison, values averaged over all grid cells within the whole city domain were also calculated for Beijing and Shanghai, as shown in Figure R1-5. The results show no obvious difference.

[Figure]

**Figure R1-3.** Diurnal variation of the mean atmospheric age and the mass concentrations of EC, $SO_4^{2-}$, $NO_3^-$, POA, SOA, and $NH_4^+$ in Beijing, Jinan, Nanjing, and Shanghai.

[Figure]

**Figure R1-4.** Hourly atmospheric age distribution of EC, $SO_4^{2-}$, and $NO_3^-$ in Beijing and Shanghai during this haze episode. White lines represent the average atmospheric age, and red lines (right y-axis) indicate total mass concentrations. The results were combined from simulations with age-bin updating intervals of 1, 3, 6, 8, and 12 h. The black lines in (a–c) indicate 12:00 to 18:00 LT 28 December.

[Figure]

**Figure R1-5**. Same as Figure R1-4 but using values averaged over all the grid cells within the whole city domain.

15. Line 282. YRD is not mentioned in Figures 5 and 6, do you mean "Shanghai"? The author should clarify the regional name and its exact reference throughout the paper.

**Response:** Thanks for the comments. YRD should be Shanghai here. To be clear, we've changed YRD and NCP to Shanghai and Beijing respectively in this paragraph. Also, we've added the exact reference of NCP and YRD in the first paragraph of Section 3.1 (see lines 196-197 and 209) in the revision.

16. Line 281: I think this is both true for Shanghai and Beijing and it is normal that SOA is older, since "SOA" is secondary. What needs to be explained is the reason for the opposite, why sometimes the age of "SOA" is younger than "POA" and "EC", and that could be the real deal of this study.

**Response:** Thanks for the comments. Yes, SOA is older than POA and EC in both Shanghai and Beijing. Figure R1-6 compared the simulated average atmospheric age of EC, POA, and SOA in Beijing, Jinan, Nanjing, and Shanghai. The results indicate that SOA is always older than that of POA and EC. To be clear, we've modified the corresponging discussion in the revision.

**Changes in the manuscript (Lines 298-303):** For both Beijing and Shanghai, the average age of SOA is generally larger than that of EC and POA, with a maximum average age of 55 h and 69 h, respectively. Similar to that of EC, POA, and $SO_4^{2-}$, the average age of SOA gradually increased during the accumulation stage in Beijing, and then decreased sharply on 30 December due to the sweeping effect of the strong northwesterly wind. In Shanghai, the average age of SOA increased from ~20 h to ~60 h within several hours during the regional transport.

[Figure]

**Figure R1-6.** Comparison of the average atmospheric age of EC, POA, and SOA in Beijing, Jinan, Nanjing, and Shanghai.

17. Line 297: I don't think that is a good explanation of the NO3- and NH4+ age pattern. Ammonium sulfate would have the same behaviors in this case. Maybe the difference comes from the different emission patterns between NOX and SOX. Also in Line 375 of the discussion, if SO4 is formed upwind, where is the location of this "upwind"? The author should provide the emission map of SOX as supporting information. Also, if this is the right explanation than it does not apply to the whole simulation period and domain. The claims in Line 294 should be limited to only "Shanghai" and periods like Dec. 31 and Jan. 1st.

**Response:** Thanks for the comments. The difference between the age patterns of $NO_3^-$ and $SO_4^{2-}$ mainly comes from the difference in their chemical production rates. We've modified Figure S11 (now Figure S13) to add the chemical production rates of $SO_4^{2-}$ before, during, and after regional transport (also shown as Figure R1-7 below). It can be found that the chemical production rate of $SO_4^{2-}$ is much lower than that of $NO_3^-$ during this episode. The maximum (mean) chemical production rates of $NO_3^-$ and $SO_4^{2-}$ averaged over the model domain are 1.7 (0.4) and 0.3 (0.08) mg/m$^2$/h, respectively. We do agree that the difference in the chemical production rates of $NO_3^-$ and $SO_4^{2-}$ is partly caused by the different emission patterns between NOx and SOx. To confirm this, we've added a new Figure S14 to show the emission map of NOx and $SO_2$ in the supporting information (also shown as Figure R1-8 below). From 2013–2017, with the implementation of stringent emission controls, China's anthropogenic emissions are estimated to decrease by 59% for $SO_2$ and 21% for NOx (Zheng et al., 2018). Measurements showed that $SO_4^{2-}$ concentration decreased by ~60% in Beijing during 2013–2018 due to reductions in $SO_2$ emissions, but $NO_3^-$ changed marginally (Li et al., 2020). Previous studies also pointed out that the contribution of $NO_3^-$

to PM$_{2.5}$ and the mass ratio of NO$_3^-$ to SO$_4^{2-}$ increased significantly in recent years, and NO$_3^-$ now surpassed SO$_4^{2-}$ to become the dominant SNA component in both NCP and YRD regions (Fu et al., 2020; Zhai et al., 2021; Zhou et al., 2022). The main factors contributing to NO$_3^-$ formation and driving its trend is the enhanced atmospheric oxidation capacity, inceased NH$_3$ availability due to the decreased SO$_4^{2-}$ concentrations, and the weakened total nitrate deposition (Xie et al., 2022).

Although an increased chemical production rate of SO$_4^{2-}$ can be found in the YRD regions during the regional transport (Figure R1-7), the contribution of SO$_4^{2-}$ with atmospheric age < 12 h to the total SO$_4^{2-}$ mass decreased significantly in both Nanjing (increased from 30% to 16%) and Shanghai (increased from 43% to 7%) (Figure 9 c and d). Additionally, the contribution of SO$_4^{2-}$ with atmospheric age > 96 h to the total SO$_4^{2-}$ mass increased significantly (increased from 16% to 33% in Nanjing and from 23% to 50% in Shanghai). Such a decrease in fresh particles and an increase in aged particles indicate that a large fraction of SO$_4^{2-}$ is formed in the upwind NCP region and then transported to YRD. According to the reviewer's suggestion, we've modified the discussion in Lines 375 and 294 as below.

**Changes in the manuscript:**

**(Lines 312-313)** NO$_3^-$ and NH$_4^+$ exhibited different age distributions compared to EC, POA, SO$_4^{2-}$, and SOA during the PM$_{2.5}$ growth stage in all the four cities.

**(Lines 398-399)** This indicates that SO$_4^{2-}$ is mainly formed in the upwind NCP region and then transported to YRD, while there is a large fraction of NO$_3^-$ formed locally in YRD.

[Figure]

**Figure R1-7.** Spatial distribution of the averaged column chemical production rates (×10$^3$ μg/m$^2$/h) of NO$_3^-$ and SO$_4^{2-}$ before, during, and after regional transport.

[Figure]

**Figure R1-8.** Spatial distribution of the emission rates (molecules/s/grid) of NO$_x$, SO$_2$. Numbers inset each panel are the mean emission rates averaged for different regions.

**References:**

Zheng, B., Tong, D., Li, M., et al.: Trends in China's anthropogenic emissions since 2010 as the consequence of clean air actions, Atmos. Chem. Phys., 18, 14095-14111, 10.5194/acp-18-14095-2018, 2018.

Li, X., Zhao, B., Zhou, W., et al.: Responses of gaseous sulfuric acid and particulate sulfate to reduced SO$_2$ concentration: a perspective from long-term measurements in Beijing. Sci. Total Environ. 721, 137700, 10.1016/j.scitotenv.2020.137700, 2020.

Fu, X., Wang, T., Gao, J., et al.: Persistent Heavy Winter Nitrate Pollution Driven by Increased Photochemical Oxidants in Northern China, Environ. Sci. Technol., 54, 3881-3889, 10.1021/acs.est.9b07248, 2020.

Zhai, S., Jacob, D. J., Wang, X., et al.: Control of particulate nitrate air pollution in China, Nat. Geosci., 14, 389–395, 10.1038/s41561-021-00726-z, 2021.

Zhou, M., Nie, W., Qiao, L., et al.: Elevated formation of particulate nitrate from N2O5 hydrolysis in the Yangtze River Delta region from 2011 to 2019, Geophys. Res. Lett., n/a, e2021GL097393, 10.1029/2021GL097393, 2022.

Xie, X., Hu, J., Qin, M., Guo, S., Hu, M., Wang, H., Lou, S., Li, J., Sun, J., Li, X., Sheng, L., Zhu, J., Chen, G., Yin, J., Fu, W., Huang, C., and Zhang, Y.: Modeling particulate nitrate in China: Current findings and future directions, Environ. Int., 166, 107369, 10.1016/j.envint.2022.107369, 2022.

18. Line 344: I would like to see the spatial distribution of averaged particle ages in a map for different periods on ground level and also vertically averaged.

**Response:** Thanks for the comments. We've added new Figures S18 and S19 (also shown as Figures R1-9 and R1-10) and some discussion to show the spatial distribution of averaged particle ages on ground level and also vertically averaged. The results show that the mean atmospheric age of PM$_{2.5}$ chemical components shows similar spatial patterns on ground level and vertical average, with larger values for the vertical average. Generally, the atmospheric age over the ocean is larger than that on the land surface due to low emissions. SO$_4^{2-}$ and SOA exhibited larger atmospheric age than the other PM$_{2.5}$ components. Before the regional transport, the mean atmospheric age of all the PM$_{2.5}$ components was larger in the NCP regions, especially for NO$_3^-$ and NH$_4^+$. During the regional transport, aged air mass transported from NCP to downwind YRD regions, leading to an increase in

the atmospheric age of EC, $SO_4^{2-}$, POA, and SOA in YRD. However, for $NO_3^-$ and $NH_4^+$, the atmospheric age changed slightly in YRD during and after the regional transport due to continuous local chemical formation. Please see lines 350-365 in the revision.

[Figure]

**Figure R1-9.** Spatial distribution of the mean atmospheric age of EC, $SO_4^{2-}$, $NO_3^-$, $NH_4^+$, POA, and SOA on ground level before, during, and after the regional transport. Units are hours.

[Figure]

**Figure R1-10.** Spatial distribution of the mean atmospheric age of EC, $SO_4^{2-}$, $NO_3^-$, $NH_4^+$, POA, and SOA averaged vertically before, during, and after the regional transport. Units are hours.

**RC3, Reviewer #3:**

Summary:

The concept of atmospheric age, which refers to the time that has passed since the emission or formation of an air pollutant, has not received much attention in previous studies. This manuscript attempted to addresses this gap by updating an age-resolved UCD/CIT model and using it to simulate the evolution of age distribution for different PM2.5 components during a severe regional haze episode in eastern China. The particle age information presented in this manuscript is interesting and provides a unique perspective on the formation and evolution of regional haze event, which is meaningful for future research in atmospheric chemistry and physics. Moreover, the manuscript is well written and easy to follow. I recommend to accept this manuscript after some minor revisions.

**Response:** Thanks for the recognition of our study. Below is the response to each specific comment.

Comments:
1.    Lines 67-68: How to define the atmospheric age of secondary particles? Is that based on the time since the emission of precursors or the formation of secondary aerosols?

**Response:** The atmospheric age of secondary particles is defined as the time they have been suspended in the atmosphere since they are formed.

2.    Lines 79-80: The key factor contributes to the changes in the hydrocarbon ratios is the difference in reaction rates of different hydrocarbons with oxidants such as OH radicals. Please modify this sentence to make it more clear.

**Response:** Thanks for the suggestion. We've modified this sentence as "Since the oxidation rates of these hydrocarbons by hydroxyl (OH) radicals vary widely, the hydrocarbon ratios change with photochemical aging" in the revision (lines 79-80).

3.    Lines 90-96: The author mentioned the dynamic age-bin updating algorithm used in the UCD/CIT model is based on the method used previously in the CMAQ model. Is there any difference between them? It's better to provide more details.

**Response:** Thanks for the comments. The dynamic age-bin updating algorithm is based on Zhang et al. (2019), who also used the UCD/CIT model. The dynamic age-bin updating algorithm is used to dynamically update the concentrations of particles between different age bins, which can be represented in Eq. 1 as mentioned in the Response to Reviewer #1's comment #1. The same algorithm has also been used in CMAQ to track the age distribution of primary and secondary inorganic aerosols (Ying et al., 2021). To clarify, we've modified the description in this paragraph in the revision. Please see lines 88-100.

4.    Lines 123-126: How many age bins are used? What's the time interval between different age bins? Also, What's the highest explicit age?

**Response:** A total of 9 age bins were used in this study, and the age bin updating frequency is set to 12 h. So the explicitly tracked particle age is 96 h. For particles with age > 96 h, their concentrations were assigned to the last age bin.

5.  Line 130: $\tau_i$ in equation (2) should be $\overline{\tau_i}$.

**Response:** Corrected.

6.  Lines 145-146: How to convert total emissions into different size bins? Please provide more details.

**Response:** Thanks for the comments. The UCD/CIT model uses a sectional method to represent the size distribution of aerosols. In this study, a total of 15 size bins, ranging from 0.01 to 10 µm in diameter, are used. Because the existing emission inventories, such as MEIC, only provide the total particulate emissions without their size distribution. The total particulate emissions obtained from MEIC and FINN were transformed into size-resolved emissions based on a library of primary particle source profiles measured during actual source tests (Hu et al., 2015). The fraction of total particulate emissions assigned to each bin is shown in Table S1 (also shown as Table R3-1 below). Please see lines 151-152 in the revision.

**Table R3-1.** Fractional apportionment of particle-phase emissions across the 15 size bins in the UCD/CIT model.

| Size bin | Diameter (nm) | Emission fractions | | | | |
|---|---|---|---|---|---|---|
| | | Industry | Residential | Power | Transportation | Wildfire |
| 1 | <10 | 0 | 0.0001 | 0 | 0.0046 | 0.0000 |
| 2 | 10-16 | 0 | 0.0004 | 0 | 0.0103 | 0.0007 |
| 3 | 16-25 | 0 | 0.0019 | 0 | 0.0140 | 0.0046 |
| 4 | 25-39.5 | 0.0004 | 0.0104 | 0.0004 | 0.0284 | 0.0437 |
| 5 | 39.5-63 | 0.0239 | 0.0295 | 0.0239 | 0.0839 | 0.1632 |
| 6 | 63-100 | 0.2542 | 0.0931 | 0.2542 | 0.2617 | 0.2601 |
| 7 | 100-160 | 0.5139 | 0.2366 | 0.5139 | 0.2194 | 0.2637 |
| 8 | 160-250 | 0.1902 | 0.2756 | 0.1902 | 0.0970 | 0.1017 |
| 9 | 250-395 | 0.0172 | 0.2225 | 0.0172 | 0.0732 | 0.0379 |
| 10 | 395-630 | 0.0002 | 0.0727 | 0.0002 | 0.0680 | 0.0343 |
| 11 | 630-1000 | 0 | 0.0163 | 0 | 0.0145 | 0.0308 |
| 12 | 1000-1600 | 0 | 0.0409 | 0 | 0.0305 | 0.0260 |
| 13 | 1600-2500 | 0 | 0 | 0 | 0.0325 | 0.0162 |
| 14 | 2500-3950 | 0 | 0 | 0 | 0.0312 | 0.0081 |
| 15 | 3950-10000 | 0 | 0 | 0 | 0.0310 | 0.0091 |

Reference: Hu, J., Zhang, H., Ying, Q., Chen, S. H., Vandenberghe, F., and Kleeman, M. J.: Long-term particulate matter modeling for health effect studies in California – Part 1: Model performance on temporal and spatial variations, Atmos. Chem. Phys., 15, 3445-3461, 10.5194/acp-15-3445-2015, 2015.

7.  Lines 155-156: So there are a total of 5 cases? Why not directly use the high time resolution

simulation, such as with a time interval of 1 hour?

**Response:** Yes. There are a total of 5 cases. In this study, a sectional method was used to represent the atmospheric age of particles, and the total number of age bins was fixed as 9 in the model. Thus, this method can only represent a limited range of particle age. For example, the simulation with a time interval of 12 h can explicitly represent particles within 96 h (12×8), and particles older than 96 h are assigned to the last age bin. Note that with an updating interval of 12 h, particles with ages of 0-12 h (12-24h, 24-36h, etc.) are assigned to the same age bin. However, in urban areas, some particles may have been emitted or formed just several hours ago due to intensive anthropogenic emissions. To get relatively detailed age information, we ran 5 cases with updating intervals of 1, 3, 6, 8, and 12 h, and then combined them together. Although the 1-hour simulation can provide the highest age resolution, the explicitly represented particle age is only 8 hours. Thus, it's not suitable to study the age distribution of particles older than 8 hours.

8.    Line 191: Figure S1 should be Figure S2. So as Figure S2 in Line 196 and Figure S3 in Line 201. The authors should check this throughout the manuscript.

**Response:** Thanks for the comments. All the figure citations have been checked and revised.

9.    Line 229: How to convert the measured OC to OM? Is that based on an assumption of OM/OC ratio? Please clarify.

**Response:** Yes. The measured OC is converted to OM by multiplying a factor of 1.64. The factor used in this study is obtained from Tan et al. (2018), which is based on measurements with an aerosol mass spectrometer. We've added this in the revision (Lines 182-183).

Reference: Tan, T., Hu, M., Li, M., et al.: New insight into $PM_{2.5}$ pollution patterns in Beijing based on one-year measurement of chemical compositions, Sci. Total Environ., 621, 734-743, 10.1016/j.scitotenv.2017.11.208, 2018.

10.   Figure 3: What time period does the result shown in Figure 3e correspond to? Additionally, the colors used for $SO_4^{2-}$ and $NH_4^+$ are too similar, making it difficult to differentiate between them. Please change.

**Response:** Thanks for the comments. Figure 3e shows the IMR (incremental mass ratio) of the major PM2.5 chemical compositions, which is calculated based on the changes in the mass concentration of particles during the $PM_{2.5}$ growth stage. Thus, the time periods are different for different cities. For Beijing and Jinan, $PM_{2.5}$ concentration began to increase on Dec. 25 and reached its peak on Dec. 29, so the corresponding time period is Dec.25–29. For Nanjing and Shanghai, the corresponding time periods are Dec. 27–31 and Dec. 28–31, respectively. The colors in Figure 3 have been modified (see Figure R3-1 below).

[Figure]

**Figure R3-1**. Mass concentrations and fractions **(a–d)** and incremental mass ratio (e) of the major PM$_{2.5}$ chemical compositions (SO$_4^{2-}$, NO$_3^-$, NH$_4^+$, EC, and OM) in Beijing, Jinan, Nanjing, and Shanghai.

11. Lines 316-317: I cannot find a "more uniform" vertical distribution for the period of accumulation stage from Figure 7. On the contrary, during the regional transport stage, the vertical distribution in the NCP region appears to be more uniform. This is because the air pollutants ahead of the cold front was lifted from the ground to the upper level.

**Response:** Thanks for the comments. From Figure 7, a "more uniform" vertical distribution in the NCP can be found during the accumulation stage when compared to that in the YRD. To be clear, we modified this sentence to "Compared to YRD, the vertical distribution of particle ages was more uniform in the NCP due to the accumulation of aged pollutants in the atmosphere under stable weather conditions" in the revision. We do agree that the vertical distribution in the NCP region appears to be more uniform during the regional transport stage when compared to that during the accumulation stage. We've added some more discussion in the revision (please see lines 333-339).

12. Lines 335-336: What is "fresh" particle? In Line 272, the author mentioned fresh particles refer to particles with atmospheric age < 24 h, while they are particles with an age of less than 12 h here. Please clarify.

**Response:** Thanks for the comments. Fresh particles refer to particles with a low atmospheric age. In this study, we defined fresh particles as particles with atmospheric age < 24 h. To be clear, we added a definition in the revision. Please see lines 275-276. Also, in line 357, we've removed the word "fresh" to avoid misunderstanding.

13. Line 375: It should be "NO$_3^-$ formed locally in YRD".

**Response:** Corrected.

14. Mistake in the reference in Line 558.

**Response:** Corrected.